# Toward seamless hydrologic predictions across spatial scales

Luis Samaniego[1], Rohini Kumar[1], Stephan Thober[1], Oldrich Rakovec[1], Matthias Zink[1],
Niko Wanders[2,7], Stephanie Eisner[3,4], Hannes Müller Schmied[5,6], Edwin H. Sutanudjaja[7],
Kirsten Warrach-Sagi[8], and Sabine Attinger[1]

[1]Department of Computational Hydrosystems, UFZ-Helmholtz Centre for Environmental Research, Leipzig, Germany
[2]Department of Civil and Environmental Engineering, Princeton University, USA
[3]Center for Environmental Systems Research, University of Kassel, Kassel, Germany
[4]Now at Norwegian Institute of Bioeconomy Research, Ås, Norway
[5]Institute of Physical Geography, Goethe-University Frankfurt, Frankfurt, Germany
[6]Senckenberg Biodiversity and Climate Research Centre (BiK-F), Frankfurt, Germany
[7]Universiteit Utrecht, Department of Physical Geography, Utrecht, The Netherlands
[8]Institute of Physics and Meteorology, University of Hohenheim, Stuttgart, Germany

*Correspondence to:* L. Samaniego (luis.samaniego@ufz.de)

**Abstract.** Land surface and hydrologic models (LSM/HM) are used at diverse spatial resolutions ranging from catchment-scale (1-10 km) to global-scale (over 50 km) applications. Applying the same model structure at different spatial scales requires that the model estimates similar fluxes independent of the chosen resolution, i.e., fulfills a flux-matching condition across scales. An analysis of state-of-the-art LSMs and HMs reveals that most do not have consistent hydrologic parameter fields. Multiple experiments with the mHM, Noah-MP, PCR-GLOBWB and WaterGAP models demonstrate the pitfalls of deficient parameterization practices currently used in most operational models, which are insufficient to satisfy the flux-matching condition. These examples demonstrate that J. Dooge's 1982 statement on the unsolved problem of parameterization in these models remains true. Based on a review of existing parameter regionalization techniques, we postulate that the multiscale parameter regionalization (MPR) technique offers a practical and robust method that provides consistent (seamless) parameter and flux fields across scales. Herein, we develop a general model protocol to describe how MPR can be applied to a particular model, and present an example application using the PCR-GLOBWB model. Finally, we discuss potential advantages and limitations of MPR in obtaining the seamless prediction of hydrological fluxes and states across spatial scales.

## 1 Introduction

> ... *"If it disagrees with experiment, it's wrong". Richard P. Feynman*

Land surface and hydrologic models (LSM/HM) are currently used at diverse spatial resolutions ranging from 1-10 km in catchment-scale impact analysis and forecasting (Christensen and Lettenmaier, 2007; Addor et al., 2014) to over 50 km in global-scale climate change simulations to estimate land surface boundary conditions of key state variables (Haddeland et al., 2011; Bierkens, 2015; Wanders and Wada, 2015). The fundamental conditions behind the applicability of the same LSM/HM model structure at different spatial scales requires that the model parameterizations are scale-invariant and that the model

estimates similar fluxes across a range of spatial resolutions. In other words, it must fulfill the flux-matching condition across scales so that the mass conservation principle can be ensured (Wood, 1997).

A parameterization is a simplified and idealized representation of sub-grid physical phenomenon that is either "too small, too brief, too complex, or too poorly understood" to be explicitly represented by a model at a given resolution (Edwards, 2010). Parameterizations require variables called predictors, effective parameters and constants also called transfer-, global- or super-parameters (Pokhrel and Gupta, 2010). Superparameters are often parameters in empirical relationships that have been found with measurements in the field or in the laboratory, e.g., regression parameters in pedo-transfer functions (Cosby et al., 1984). They are often tuned to represent observed variables and often have no physical meaning. These parameters constitute simplified surrogates to compensate for the missing sub-grid processes that are not accounted for within a modeling system (Brynjarsdottir and O'Hagan, 2014).

Effective parameters of LSMs/HMs are usually obtained by ad-hoc procedures (e.g., automatic calibration) at a given spatial resolution for a given modelling domain. As a consequence of this standard practice, parameter fields of LSMs/HMs often exhibit artificial spatial "discontinuities" such as calibration imprints circumscribing river basins boundaries, and consequently they are not seamless (Merz and Blöschl, 2004; Li et al., 2012). Inconsistent patterns of effective parameter fields for land surface geophysical properties across spatial scales constitute a clear indication that their parameterizations are not scale-invariant. There are several reasons explaining this parameterization deficiency. With the advent of electronic computers, the performance of general circulation models (GCMs), numerical weather prediction (NWP) models (Pielke Sr, 2013), land surface models (Liang et al., 1994; Sellers et al., 1997; Niu et al., 2011), and hydrologic models (Batjes, 1996; Lindstrom et al., 1997; van Beek et al., 2011; Samaniego et al., 2010b) has been increased mainly by improving model conceptualization (i.e., the number of process descriptions) and/or spatial resolution since the storage capacity and computational power allowed for it (Le Treut et al., 2007; Wood et al., 2011; Bierkens et al., 2014). As a result, parameterizations in LSMs have also increased in their complexity during the past decades (Sellers et al., 1997; Fisher et al., 2014). The procedures to estimate effective parameters required for the parameterizations, however, remained unchanged. For example, LSMs evolved from simple aerodynamic bulk transfer schemes with uniform description of surface parameters during the 1970s, to detailed LSMs having consistent description of the exchange of energy and matter between the atmosphere, the vegetation, and the land surface (Sellers et al., 1997). State-of-the-art LSMs, such as the Community Land Model version 4 (Bonan et al., 2011) and Noah-MP (Niu et al., 2011), however, still use quite simple pedotransfer-functions based on work of Clapp and Hornberger (1978) and Cosby et al. (1984) to estimate fundamental soil properties such as porosity (Oleson et al., 2013).

Further reasons that have prevented the improvement of parameterization techniques are: 1) the lack of procedures and theories for linking physical properties (e.g., soil porosity) that can be measured at the field scale with "effective" parameter values that represent the aggregate behavior of the land characteristics at the scale of a grid cell required in LSMs or HMs. 2) Poor understanding of the scaling of parameters (Dooge, 1982) and its influence on the hydrological response of the system (Wood, 1997; Wood et al., 1988). 3) Limited inclusion of sub-grid heterogeneity in hydrological parameterizations and multi-scale modeling of hydrologically relevant variables as suggested by Famiglietti and Wood (1995, 1994); Liang et al. (1996). 4) Lack of significant progress on the applicability of seminal upscaling theories (Miller and Miller, 1956; Dagan, 1989; Gelhar,

1993; Neuman, 2010; Kitanidis and Vomvoris, 2010) developed for sub-surface hydrologic problems into LSM/HMs. And 5) lack of transparency in most of the existing LSM/HM source codes with respect to the meaning, origin and uncertainty associated with the hard-coded numerical values (i.e., parameters) either in the code or in the look-up tables (Mendoza et al., 2015; Cuntz et al., 2016).

Consequently, it is possible to assert that model parameterization is an old, ubiquitous, and recurring problem in land surface and hydrologic modeling. Considering this lack of coherent development during the past decades, we can still concur with Dooge (1982, p.269) and say that the "parameterization of hydrologic processes to the grid scale of general circulation models is a problem that has not been approached, let alone solved."

There are potential methods available in the literature that may lead toward coherent parameterizations and prediction of
water and energy fluxes in LSMs/HMs. For example: 1) sidestepping the scaling problem of key model parameters by assuming scale-independent distribution functions with regionalized distribution parameters (Intsiful and Kunstmann, 2008), 2) find strong links between model parameters to mapped geophysical attributes via regularization procedures (Pokhrel and Gupta, 2010), and 3) find strong links between of observed functional responses of hydrological systems and geophysical characteristics (Yadav et al., 2007). These methods, however, alone may not satisfy the flux-matching criteria.

In contrast with these existing methods, we argue that the multiscale parameter parameterization (MPR) technique (Samaniego et al., 2010b) offers a framework to link the field scale (observations) with the catchment scale (Dooge, 1982). MPR also accounts for the effect of the spatial variability and non-linearity of geophysical characteristics in the parameterization of hydrologic processes that operate at a range of spatial resolutions (Dooge, 1982; Wood et al., 1988). Depending on the conditions imposed to the parameter estimation technique, MPR can lead to parameterizations that satisfy the flux-matching criteria and
hence contributes to obtain seamless parameter and water flux fields. Because MPR relies on empirical transfer functions and upscaling operators to link geophysical properties with model parameters, it provides a very effective procedure to transfer "global parameters" to scales and locations other than those used in calibration (Samaniego et al., 2010a, b; Kumar et al., 2013b). This dependency on several transferable coefficients also contributes to minimize a serious drawback of spatially explicit models called "over-parametrization" (Beven, 1995).

In this study, we analyze to which extend existing LSM/HM parameterizations are limited to obtain seamless predictions of water fluxes and states across multiple spatial resolutions. Through several modeling experiments addressing Wood (1990)'s query (i.e., "What modeling experiments need to be performed to resolve the scale question ..."), we demonstrate that a large portion of the predictive uncertainty in existing LSM/HMs originates from the deficient estimation of effective parameters, which leads to a lack of scale invariance and thus to their poor transferability across scales and locations. These experiments
also aim to help the modeler to reveal poor performing parameterizations, i.e., those that exhibit non-seamless fields. Finally, based on our past experiences and aiming to address the challenges stated above, we develop a protocol that systematizes the application of the MPR technique for any LSM/HM and demonstrate its effectiveness by implementing it into the PCR-GLOBWB model.

## 2 Current parameterization techniques

### 2.1 The state-of-the-art

The most common parameterization techniques found in literature are: 1) look-up-tables (LUT), 2) manual or automatic calibration, 3) hydrologic response units (HRU), 4) representative elementary watersheds (REW), 5) a priori regularization functions,
6) simultaneous regionalization/regularization functions, 7) dissimilarity-based metrics to transfer model parameters.

The simplest technique to assign a parameter value to a modeling unit (e.g., grid cell, HRU, sub-catchment) is based on a LUT. In this case, a categorical index associated with a modeling unit links it with information taken from an external reference file (i.e., the LUT) which maps this index with parameter values that are usually taken from the literature. This technique is commonly used in most of the (operational) LSMs such as CABLE, CHTESSEL, CLM, JULES, Noah-MP (Kowalczyk et al.,
2006; Viterbo and Beljaars, 1995; ECMWF, 2016; Oleson et al., 2013; Best et al., 2011; Niu, 2011). A disadvantage of this method is the difficulty to perform sensitivity analysis (Cuntz et al., 2016). Moreover, the number of classes defined in LUT is often limited to a few (e.g., 13 soil classes in Noah-MP) resulting in non-seamless parameter fields that are not continuous.

Manual or automatic calibration is a commonly used technique to parameterize spatially lumped hydrologic models (e.g., Crawford and Linsley, 1966; Burnash et al., 1973; Lindstrom et al., 1997; Edijatno et al., 1999; Fenicia et al., 2011; Martina
et al., 2011; Andréassian et al., 2014; Singh et al., 2014) and semi-distributed hydrologic models (e.g., Leavesley et al., 1983; Kavetski et al., 2003; Lindström et al., 2010; Hundecha and Bárdossy, 2004; Merz and Blöschl, 2004; Hundecha et al., 2016). The aim is to minimize the disagreement between model simulations and observations. In the majority of the cases, the target variable is streamflow. The main drawback of this parameterization technique is that the parameter fields, which are obtained by colocating lumped model parameters from sub-basins, are doubtful because they exhibit sharp discontinuities along indi-
vidually calibrated sub-basin boundaries despite having spatial continuity in basin physical attributes like soil, vegetation and geological properties that govern spatial dynamics of hydrological processes (Merz and Blöschl, 2004; Li et al., 2012; Blöschl et al., 2013). In addition, the "patchwork quilt" parameter fields shown in these references exhibit significant sensitivity to the calibration conditions as demonstrated by Merz and Blöschl (2004). Thus, models that are parameterized with this technique may exhibit (1) poor predictability of state variables and fluxes at locations and periods not considered in calibration and (2)
sharp discontinuities along sub-basin boundaries in state, flux and parameter fields (e.g., Merz and Blöschl, 2004; Lindström et al., 2010). Parameter fields derived from basin-wise "calibrated" lumped models lack spatial seamlessness, and thus are "inadequate representations of real-world systems" (Savenije and Hrachowitz, 2017). Moreover, excessive reliance on parameter calibration leads to deficient performance at interior points of the basin or at other locations at which the model was not calibrated (Pokhrel and Gupta, 2010; Lerat et al., 2012; Brynjarsdottir and O'Hagan, 2014).
There have been many attempts to improve the parameterization of lumped and semi-distributed models by further discretizing the sub-basins into a given number of regions that exhibit nearly similar hydrologic behavior, i.e., the so-called hydrologic response units (HRU) concept initially proposed by Leavesley et al. (1983) and further developed by others (e.g., Flügel, 1995; Beldring et al., 2003; Blöschl et al., 2008; Viviroli et al., 2009; Zehe et al., 2014). Unfortunately, results obtained in these parameterization attempts have not been very successful in realistically representing the spatial variability of model parame-

ters, states and fluxes because of the lack of regionalized parameters and the unabridged reliance on parameter calibration to improve model performance (Kumar et al., 2010). Commonly, the effective parameters estimated for the HRUs are found by automatic calibration. Efforts have been made to enforce continuity on parameter fields (Gotzinger and Bárdossy, 2007; Singh et al., 2012), but with somewhat limited success during the transferability of parameters across scales and locations. In addition, models parameterized using HRUs do not lead to mass conservation of water fluxes (i.e., flux-matching) when applied to scales other than those used for calibration (Kumar et al., 2010, 2013b). Recent attempts have been made to improve the HRU concept to increase the seamless representation of parameters, states and fluxes (Chaney et al., 2016a). However, this concept has not been tested for scalability and seamlessness of the estimated fields at coarse resolutions. Lately, a thermodynamic reinterpretation of the HRU concept was proposed by Zehe et al. (2014), but to date, the implementation of this approach has not found its way into meso- to macro-scale LSMs/HMs.

The representative elementary watershed approach (Reggiani et al., 1998) is an interesting theoretical concept, which scales mass and momentum balance equations. Unfortunately, to the best of our knowledge, it has not been used to estimate effective parameters at meso- and regional scales.

A priori regularization functions (e.g., pedo-transfer functions) were introduced by Koren et al. (2013) to ensure the "inappropriate randomness in the spatial patterns of model parameters", i.e., the lack of seamlessness. Unfortunately, in this case, the parameters (or coefficients) of regularization functions were not subject to parameter estimation or to the verification of their ability to predict fluxes and states across various scales. The use of empirical point-scale-based relationships to link geophysical characteristics with LSM/HM parameters and the assumption that their coefficients are universally applicable with certainty (e.g., the coefficients in the Clapp and Hornberger (1978) pedo-transfer functions) are the major reasons for the proliferation of hidden parameters in LSM/HM code (Mendoza et al., 2015; Cuntz et al., 2016). It is of pivotal importance to understand that these point-scale relationships should not be applied beyond the scale at which they were derived.

Many types of regionalization (or regularization) approaches have been tested for semi-distributed and distributed models. According to Samaniego et al. (2010b), these approaches can be broadly classified into post-regionalization and simultaneous regionalization approaches, depending on if the regionalization function parameters (or global parameters) are estimated after (Abdulla and Lettenmaier, 1997; Seibert, 1999; Wagener and Wheater, 2006; Livneh and Lettenmaier, 2013) or during the model calibration (Fernandez et al., 2000; Hundecha and Bárdossy, 2004; Gotzinger and Bárdossy, 2007; Pokhrel and Gupta, 2010). None of these procedures consider the sub-grid variability of the model parameters or geophysical characteristics. Livneh and Lettenmaier (2013) noted that most of these regionalization procedures exhibit limited transferability because of the use of discrete soil texture classes as predictors, and very likely discontinuous parameter fields.

Recently, a dissimilarity-based regionalization technique was used by Beck et al. (2016) to generate an ensemble of global parameters of the HBV model at a 0.5° resolution for global-scale hydrological modeling. A shortcoming of this approach is the use of ad hoc nearest-neighbor interpolation of parameter fields to fill gaps where no donor basins are available in (geographically) surrounding regions. Following a similar concept of that of Beck et al. (2016), the parameterization method proposed by Bock et al. (2016) for the Contiguous United States (CONUS) will likely lead to discontinuous parameter fields for reasons similar to those mentioned above.

Many attempts have been made in the land surface modeling community to address Dooge's challenges, especially with respect to the transferability of model parameters across locations and scales and to obtain seamless parameter fields. One of the earliest prominent experiments was conducted in the Project for Intercomparison of Land-surface Parameterizations (PILPS) (Wood et al., 1998). In this project, calibrated LSM parameters were transferred from small catchments to their nearest computational grid cells. The results indicated that LSMs exhibited poor transferability across space, leading to significant differences in the partitioning of water and energy fluxes. For instance, Troy et al. (2008) used calibrated VIC model parameters from small basins to generate parameter fields for continental-scale land surface modeling by "linearly interpolating to fill in those grid cell not calibrated" on a sparse grid. As noted by Samaniego et al. (2010b), this type of regionalization is inadequate because of the nonlinearity of soil and geological formations. The spatial patterns of model parameters that would be obtained by ad-hoc extrapolations based on calibrated parameters from small basins or grid cells would most likely lead to unrealistic parameter fields with spatial discontinuities circumscribing river basins, as shown in recent studies by Wood and Mizukami (2014) and Mizukami et al. (2017) for the VIC model parameters.

Recent community-driven efforts, such as the Protocol for the Analysis of Land Surface Models (PALS) and the Land Surface Model Benchmarking Evaluation Project (PLUMBER) (Haughton et al., 2016), indicate that the hurdles noted in PILPS have not been overcome. Thus, it is required to gain understanding on whether the inferior predictability of many LSMs evaluated with empirical benchmarks in the PLUMBER project (e.g., CABLE, CHTESSEL, JULES, Noah) may be the result of deficient parameterizations, among other factors.

## 2.2 Parameterization of soil porosity and available water capacity in selected LSMs/HMs

Above mentioned challenges that we face in estimating key physical parameters in LSM/HMs has been intensively discussed in many studies (Gupta et al., 2014; Bierkens et al., 2014; Bierkens, 2015; Clark et al., 2016, 2017; Mizukami et al., 2017; Peters-Lidard et al., 2017). To further visualize the problems and to understand the deficiencies of current parameterization techniques, we selected a representative sample of LSMs/HMs used for research and/or operational purposes, namely: CABLE, CLM, JULES, LISFLOOD, Noah-MP, mHM, PCR-GLOBWB, WaterGAP2 (30 arcmin), WaterGAP3 (5 arcmin), CHTESSEL, and HBV. These models vary in process complexity and spatial resolution.

We selected soil porosity as an example to visualize existing shortcomings because it is one of the most common parameters in many LSMs/HMs. This parameter controls the dynamic of several state variables and fluxes such as soil moisture, latent heat, and soil temperature, and its sensitivity has been demonstrated in various studies (Goehler et al., 2013; Cuntz et al., 2015; Mendoza et al., 2015; Cuntz et al., 2016). A representation of the porosity of the top 2 m soil column in these models over the Pan-EU is shown in Figure 1. The Pan-EU domain was selected for depiction, but we note that the problem is general and persistent across other domains (Mizukami et al., 2017). For cases in which a HM does not use this parameter, the "available water capacity" (WaterGAP) or the "field capacity" (HBV) were selected as a surrogate due to their similarity with porosity. Both surrogate fields are normalized (in space) to ease their comparison with the porosity fields. Soil porosity is expressed in $m^3 m^{-3}$ to ease the comparison among different models.

The following lessons can be learned from Figure 1: 1) there is a large variability in the parameterization of this key physical parameter because none of the analyzed models have comparable spatial patterns or comparable estimates at a given location. It should be noted that the definition of the selected parameter is rather simple: it represents the ratio of the volume of voids to the total volume in the soil column. One can now wonder how large the uncertainty of other parameters would be (e.g., hydraulic conductivity) whose relationship with soil properties is very nonlinear. 2) The degree of seamlessness strongly depends on the level of aggregation and the upscaling of underlying soil texture fields. For example, porosity for WaterGAP is substantially different in spatial pattern and magnitude for 30arcmin and 5arcmin simulations. On the contrary, the spatial pattern and magnitude for porosity used in mHM remains almost unchanged for application at 30 arcmin and 5 arcmin resolution. 3) A parameter field becomes highly discontinuous and patchy when, for a given model, the parameter is calibrated in a limited domain (or basins) and then extrapolated to other regions (e.g., as shown in the panel corresponding to the HBV). 4) These experimental results confirm the postulation of Dooge (1982) that the parameterization of the existing state-of-the-art LSM/HMs at large and continental scales is *still* an unsolved problem.

The analysis of current parameterization techniques allow us to put forward the following questions: (1) Why are there such large differences between models in estimating a parameter that has a physical meaning? (2) What are the consequences of poor parameterizations on the spatio-temporal dynamics of state variables and fluxes? (3) What are the consequences of model calibration on parameter fields? (4) Are current model parameterizations scale invariant? (5) Do the fluxes estimated with these models at various scales satisfy the fundamental mass conservation criterion (hereafter denoted the flux-matching test)?

## 3 Seamless parameterization framework

### 3.1 The flux-matching postulation

The key postulation aiming at obtaining scalable (global) parameters that are transferable across locations and scales was proposed by Samaniego et al. (2010b) and further tested in Kumar et al. (2013b, a) and Rakovec et al. (2016b). We hypothesize that

**Flux matching** across scales leads to quasi scale-invariant global parameters $\hat{\gamma}$, thus:

$$\sum_i \sum_t \left| W_i(\hat{\gamma},t)a_i - \sum_{k \in i} w_k(\hat{\gamma},t)a_k \right| \rightarrow 0, \quad \forall i \in \Omega. \tag{1}$$

Here, $k$ denotes the sub-grid elements constituting a given modeling cell $i$ with area $a_k$. $i$ denotes a modeling grid cell $i$ with area $a_i$. $W_i$ and $w_k$ denote fluxes at two modeling scales $\ell_1$ and $\ell'_1$, respectively, with $\left(\frac{\ell_1}{\ell'_1}\right)^2 = \frac{a_i}{a_k}$. $\Omega$ denotes the modeling domain, e.g., a river basin, and $t$ a point in time. It should be noted that the topology of the cells at either level is not specified. Normally, rectangular grid cells are used for convenience, but this is not a necessary condition. This strong flux-matching condition can be used as a penalty function or as an additional test to discriminate parameter sets obtained with conventional parameter estimation approaches.

## 3.2 The MPR approach

Multiscale parameter regionalization (MPR), proposed by Samaniego et al. (2010b), aims to estimate model parameters that are seamless across scales, satisfy the flux-matching conditions (see Section 3.1), and enable the transferability of global or transfer-function parameters across scales and locations (Samaniego et al., 2010a, b; Kumar et al., 2013a; Wöhling et al., 2013; Livneh et al., 2015; Rakovec et al., 2016a). The development of MPR is ongoing work. Regionalization functions used in MPR for the mHM model (www.ufz.de/mhm) by Samaniego et al. (2010a) were further improved by Kumar et al. (2013b). More recently, a model-agnostic implementation of MPR has been proposed by Mizukami et al. (2017) and tested in the VIC model in over 500+ basins in the CONUS. The study of Mizukami et al. (2017), in contrast to the present study, does not include flux-matching tests nor the evaluation of model skill across different spatial scales.

The scaling problem in MPR is addressed by using process specific representative elementary areas (REA) that determine the minimum computational grid size $\ell_1$ at which the continuum assumptions can be used without explicit knowledge of the actual patterns of the topography, soil, or rainfall fields (Wood et al., 1988). The REA of a specific process, such as streamflow, can be determined by conducting a careful sensitivity analysis as shown by Samaniego et al. (2010b). To estimate an "effective" model parameter (e.g., total soil porosity) at the selected modeling scale, it is first necessary to estimate its variability at a much finer scale $\ell_0 \ll \ell_1$ such that the effects of its spatial heterogeneity can be adequately represented. In other words, the parameter at the fine scale $\ell_0$ represents the minimum support at which the proposed equations are still valid. Barrios and Francés (2011) indicated that a suitable estimate of $\ell_0$ for a given parameter could be near its correlation length. The sub-grid variability of a parameter $\boldsymbol{\beta}_0$ depends, in turn, on the spatial heterogeneity of geo- and bio-physical characteristics ($\mathbf{u}_0$), such as terrain elevation, slope and aspect, soil texture, geological formation, and land cover, which are now available at hyper-resolution for the entire globe. The mathematical relationships that link model parameters with these characteristics at the finer resolution are called pedo-transfer, regionalization or regularization functions $f$ (Clapp and Hornberger, 1978; Cosby et al., 1984; Wösten et al., 2001). The constants required in these functions are usually denoted as global parameters $\hat{\boldsymbol{\gamma}}$, thus $\boldsymbol{\beta}_0 = f(\mathbf{u}_0, \hat{\boldsymbol{\gamma}})$. Note that the fields $\boldsymbol{\beta}_0$ and $\mathbf{u}_0$ are dependent on space and time, but the vector $\hat{\boldsymbol{\gamma}}$ is not.

Regularization functions are commonly used in mathematics and statistics to solve ill-posed problems (which is the case when the parameters of a distributed LSM/HM are determined by calibration) and/or to prevent over-fitting. The direct consequence of the regularization is the substantial decrease in degrees of freedom of the optimization problem because the cardinality of the gridded parameter fields $\#\{\boldsymbol{\beta}_0\}$ is orders of magnitude larger than that of the vector of the global parameters $\#\{\hat{\boldsymbol{\gamma}}\}$. Hence, MPR is a parsimonious parameterization technique that offers spatially continuous parameter fields and removes spatial discontinuities in water fluxes and states, as observed by Gotzinger and Bárdossy (2007) and discussed by Mizukami et al. (2017). From the Bayesian point of view, the regularization functions impose a prior distribution on the model parameters. Consequently, greater care should be taken in their selection.

The second step of the MPR approach consists of upscaling the sub-grid distribution of a regionalized parameter to the modeling scale. In other words, $\boldsymbol{\beta}_1 = \langle \boldsymbol{\beta}_0 \rangle$. Here, the symbol $\langle \cdot \rangle$ represents an averaging or scaling operator that is parameter-

specific, and thus $\beta_1$ denotes the upscaled effective parameter field. It is important to note that this scaling operator is not necessarily the arithmetic mean.

A schematic representation of the MPR procedure can be seen in Figure 2. In short, the motto of MPR is "estimate first, then average" whereas other existing regionalization methods follow the opposite approach of "average first, then estimate."

Because the processes in LSM/HMs are highly nonlinear, this sequence of operations does not commute. The consequences can be dramatic (to be shown in the results section). The latter, which is the standard approach, does not preserve fluxes/states across scales, whereas MPR does to a considerable extent. The key question here is in finding the right scaling rule for the model parameters such that the fluxes/states are preserved across a range of spatial scales.

Model parameters at the $\ell_1$ scale (i.e., 1 km to 100 km) are called "effective" parameters because they cannot be measured

by physical means at this resolution and can only be inferred by heuristic relationships $f(\cdot)$. Thus, it is essential that the inequality $\ell_0 \ll \ell_1$ is fulfilled so that the law of large numbers leads to stable estimates of the effective parameter $\beta_1$ having low uncertainty. Since every LSM/HM (e.g., those mentioned in Section 2) contains "effective" model parameters, depending on heuristic relationships (that are hidden in the source code in many cases (Mendoza et al., 2015; Cuntz et al., 2016)), it is logical that existing LSM/HMs are subject to parameter uncertainty. These models can be treated as stochastic models,

even though their governing equations are deterministic in nature and based on physical principles such as the conservation of mass and energy (Clark et al., 2015; Nearing et al., 2016). Effective parameters should not be the pure result of a blind calibration algorithm. MPR varies from other regionalization approaches in that the introduced relationships may lead to seamless parameter fields and model simulations fulfilling the flux-matching condition.

Currently, MPR is the only method that consistently and simultaneously addresses the scale, nonlinearity and over-para-

meterization issues if global parameters are estimated simultaneously at multiple locations (i.e., basins). The MPR approach also addresses the principle of scale-dependent subgrid parameterization (i.e., "net fluxes must satisfy the conservation of mass" proposed by Beven (1995)) but does not adhere to Beven's other principles, such as that sub-grid parameterizations may be data- and scale-dependent (principle 3 and 4 in Beven (1995)), because exhaustive tests reported in the above mentioned references carried out over hundreds of river basins do not appear to support them. We find MPR to be a robust technique that

has the ability to provide "effective parameters" and is capable of addressing the scaling problem; in this sense, it diverges from the Beven's view (Beven, 1995, p.507) that these "effective parameters" are an "inadequate approach to the scale problem". Furthermore, MPR differs on the regionalization and aggregation scheme (i.e., patch model areal weighting) proposed by Beven (1995, p.520).

The selection of regionalization functions and scaling operators is fundamental to ensure the transferability of global pa-

rameters across scales and to guarantee the seamlessness of parameter fields across scales, e.g., from $\ell_1$ to $2\ell_1$ ... and so on. Samaniego et al. (2010b) proposed that the key to determining them is the flux-matching condition mentioned above. A seamless parameter field $\beta_1$ can be interpreted as the corollary of the flux-matching condition. Moreover, MPR employs geophysical properties at $\ell_0$ that allow for a representative sample at the hyper-resolution promoted by Wood et al. (2011) and Bierkens et al. (2014).

### 3.3 Protocol for implementing the MPR approach

The development of LSM/HMs and their parameterizations should be guided by a strict hypothesis driven framework (Nearing et al., 2016) that aims at finding parsimonious and robust parameter sets that fulfill the flux-matching condition and a number of efficiency metrics that are not used during the parameter estimation phase. A multivariate, multiscale evaluation assessing the reliability of model simulations should follow the scheme presented in Rakovec et al. (2016a). Based on our previous experiences, we synthesize a formalized scheme (i.e., protocol) for systematically implementing the MPR technique in other LSMs/HMs with the aim to obtain a robust and seamless parameterization. A graphical depiction of the estimation procedure at multiple scales is shown in Figure 2.

1. Retrofit the source code of an LSM/HM so that all model parameters are exposed to analysis algorithms. Parameters are the values of a model that can be considered random variables, i.e., those that are subject to various outcomes and can be fully defined by a probability density function. Parameters should not be confused with numerical or physical constants.

2. Determine a set of the most sensitive model parameters through a sensitivity analysis (SA). For computationally expensive LSMs such as CLM or Noah-MP, computationally frugal methods such as the elementary Effects method (Morris, 1991), its enhanced version such as that proposed by Cuntz et al. (2015), or the distributed evaluation of local sensitivity analysis (DELSA, Rakovec et al., 2014; Mendoza et al., 2015) are of particular interest because use of the popular standard Sobol' method (Sobol', 2001) can be computationally expensive although still possible (Cuntz et al., 2016).

3. Regionalize sensitive model parameters that exhibit marked spatial variabilities. The selection of the regionalization function $f(\cdot)$ can be guided by existing literature or by step-wise methods (e.g., Samaniego and Bárdossy, 2005). This regularization step should be conducted at the highest available spatial resolution for all predictor fields. This resolution is denoted as level $\ell_0$. The output of the regularization is the parameter field $\boldsymbol{\beta}_0$.

4. Estimate effective parameter fields $\boldsymbol{\beta}_1$ using upscaling operators based on the underlying sub-grid variability $\boldsymbol{\beta}_0$. The scale $\ell_1$ is determined by synthetic experiments aimed at finding the optimal REA for processes related to the parameter in question (Samaniego et al., 2010b; Kumar et al., 2013b).

5. Estimate the global-parameters $\hat{\boldsymbol{\gamma}}$ using standard optimization algorithms (simulated annealing, shuffled complex evolution (SCE), dynamically dimensioned search (DDS)) by minimizing a compromise metric that includes observations at multiple scales and locations (Duckstein and Opricovic, 1980; Rakovec et al., 2016a). The compromise metric could also include hydrologic signatures to extract as much information from a time series as possible (Nijzink et al., 2016).

6. Perform multi-basin, multi-scale, multi-variate cross-validation tests to evaluate the robustness of the regionalization functions, scaling operators, and global parameters (Rakovec et al., 2016a).

7. If the cross-validation tests provide satisfactory results (e.g., Kling-Gupta efficiency (KGE) of the compromise solution $> 0.6$), then evaluate the flux-matching condition given by eq. 1. If the total error is too large to be tolerated, repeat steps 3 to 8.

8. Evaluate the parameter seamlessness and the preservation of the statistical moments of fluxes and states across scales (seamless prediction step in Figure 2).

It should be noted that any of the steps above can be tested within a sequential hypothesis testing framework (Clark et al., 2016). A substantial difference from a standard model optimization exercise is that the transfer function $f(\cdot)$ (step 3) and the upscaling operator (step 4) can also be modified in the modeling protocol.

Failure to satisfy the imposed condition, such as the flux-matching test, after exhaustively testing the options in steps 3 to 6 may indicate deficits in process understanding and/or poor data. Consequently, the evaluation step should also provide guidance on detecting and separating the errors stemming from process conceptualization (modeling) and input data.

## 3.4 Seamless parameter fields across multiple scales using MPR

In Section 3.2, it was postulated that the MPR technique aims at estimating seamless parameter fields across scales which minimize the occurrence of artificial discontinuities and ease the transferability of model parameters across scales and locations. The latter has been tested and reported in many studies in Europe, USA, and other basins worldwide (Samaniego et al., 2011; Kumar et al., 2013b, a; Rakovec et al., 2016b, a). In this study, we provide evidence in favor of the former postulation.

To achieve this goal, the mHM model is parameterized using the multiscale parameter regionalization technique (MPR) (Samaniego et al., 2010b) with hyper-resolution fields of geophysical characteristics at $\ell_0 = 500$ m resolution as input. Among them, the land cover data was obtained from the Corine data sets (land.copernicus.eu/pan-european/corine-land-cover), and the soil texture information was derived from SoilGrids (soilgrids.org). These very detailed and homogenized soil texture fields provide the fractions of clay and sand, mineral bulk density, and fraction of organic matter for six soil horizons up to 2 m deep. A hyper-resolution digital elevation model (DEM) over Europe (approximately 30 m) from the GMES RDA project (EU-DEM; www.eea.europa.eu/data-and-maps/data/eu-dem) was used to derive terrain characteristics such as slope, aspect, flow direction. The underlying hydro-geological characteristics are based on the International Hydrogeological Map of Europe (IHME; www.bgr.bund.de/ihme1500), available at a 1:1 500 000 scale. Details on the pedo-transfer function used for these simulations can be found in Livneh et al. (2015). mHM global parameters were obtained by closing the water balance over selected river basins in Europe (Rakovec et al., 2016a).

Based on these settings, which constitute the basis for the EDgE project (edge.climate.copernicus.eu), we estimated porosity fields at three modeling resolutions of $\ell_1 = 5$, 10, and 25 km, based on the same $\ell_0$ support information. Following the MPR procedure depicted in Figure 2, the parameter fields for the mHM model at these three resolutions can be estimated. Results are shown in Figure 3.

The results illustrate that the MPR approach can preserve the spatial pattern of the porosity fields (see panels (a), (b) and (c) in Figure 3) and the first and second moments of its probability density function shown in panels (e)-(g). Two-sample Kolmogorov-Smirnov tests indicate that there is insufficient evidence to reject the null hypothesis that any of the three possible pairs of empirical distributions were drawn from the same unknown distribution. This highlights that the MPR approach leads

to consistent parameter fields across scales. In this case, the mean porosity is estimated to be 0.42 m3 m-3 independent of the scale.

## 3.5 Limitations of the MPR approach

The MPR approach, as any method, has some limitations. One of the crucial aspects of MPR is the selection of transfer functions and upscaling operators. Existing theories could be the first guess, but in the case that nothing is available, the protocol proposed in Section 3.3 could be used to guide the search of robust transfer functions. Testing the model parameterization for flux matching conditions across a range of basin and spatial scales may help to identify adequate upscaling operators. This procedure, although tedious, is the only solution for the moment.

In the case that some state variables change over time (e.g., land cover/use), or during parameter estimation, the MPR algorithm have to be linked to the model because every time a global parameter ($\hat{\gamma}$) is re-estimated, all related model parameters model ($\beta_1$) have to be updated as illustrated in Figure 2. The computational cost of performing MPR is therefore larger than other parameterization method discussed before.

Another limitation of the applicability of the MPR technique until recently was its availability only as an intrinsic module of the mHM model (www.ufz.de/mhm). This implies that tailored algorithms (i.e., source code) to perform the regionalization and upscaling of parameters for a target LSM/HM have to be developed from scratch, as it is demonstrated here as a case study for the PCR-GLOBWB model. This activity is of course time consuming and not pleasing due to its complexity. For this reason, Mizukami et al. (2017) have started a community effort to develop a model-agnostic MPR implementation (MPR-flex), which has been so-far evaluated for the VIC model.

The availability of high resolution bio-physical characteristics at the spatial scale $\ell_0$ constitutes another limitation of the applicability of MPR. Since the sub-grid variability is fundamental to estimate robust effective parameter values at coarser scales, the minimum scale at which a model can be applied ($\ell_1$) is strongly determined by the data availability. For example, if the soil data is available for the Pan-EU domain at $\ell_0$=250 m, the $\ell_1$ should not be lower than 1000 m, so that each modeling cell ($\ell_1$) have a representative number of underlying sub-grid cells ($\ell_0$).

MPR has been mainly developed for a hydrologic model representing the water cycle. However, land surface models also include the energy and carbon cycles and thus have greater complexity. In particular, they have more detailed representation of vegetation. It is a topic for future research to develop a MPR approach (i.e., transfer functions and upscaling operators) for plant functional type-specific parameters such as carboxylation rate and the slope of the Ball-Berry equation for stomatal conductance (Ball et al., 1987), which are required for a successful implementation of MPR in LSMs.

Finally, the computational effort for MPR is also considerably larger in comparison with other methods, because of its requirement to estimate model parameters ($\beta_0$) at the highest resolution at which the bio-physical characteristics are available. The computational time, however, could be substantially reduced by using a restart file (i.e., a data set containing a copy of all parameters, state variables, and fluxes of a model at a given point in time). If this capability is available, the MPR estimation can be greatly reduced for operational simulations because the effective parameter fields and past modeled states do not need to be estimated often.

## 4 Experiments to reveal non-seamless parameterizations

In this section we perform four modeling experiments, inspired on Wood (1990)'s recommendation, to investigate: 1) the effects of the over-calibration of global parameters on the spatial patterns of modeled state variables. 2) The effects of a parameterization technique on the spatial pattern of effective parameters. 3) The effects of a parameterization technique on the dynamics of a state variable. And, 4) the effects of not satisfying the flux-matching condition on simulated flux across different spatial scales. In these experiments four models are employed: mHM, Noah-MP, PCR-GLOBWB, and WaterGAP.

### 4.1 Effects of on-site model calibration

As noted in the introduction, on-site (basin-specific) parameter estimation based on HRU or similar techniques (such as clustering grid cells or sub-basins into regions that exhibit quasi-similar hydrological behavior) leads to non-seamless parameter fields such as those reported in Merz and Blöschl (2004). Here, we go one step further to show the consequences of this common practice on state variables such as soil moisture. Our postulation is that an on-site calibration of global-parameters $\hat{\gamma}$ leads to biased state variables even with regularization techniques such as MPR. To falsify this postulation, we performed two model simulations denoted "on-site" and "multi-site" calibration schemes. In both cases, we used the mHM setup described in Rakovec et al. (2016b) over the Pan-EU domain at a 0.25° resolution.

In the first simulation, we perform on-site calibrations at 400 river basins in the Pan-European domain. Subsequently, the respective optimized parameter sets are used in each corresponding basin to generate the target variable, in this case, the daily soil moisture of the top 1 m soil column. Lastly, daily soil moisture fields are assembled using the independent basin simulations for the entire Pan-EU domain. The results of this experiment are shown in panel (a) of Figure 4 for a day in August 2005. In the second simulation, the global-parameters $\hat{\gamma}$ are estimated simultaneously for a set of 13 basins covering various hydro-climatic regimes in the Pan-EU domain. The corresponding soil moisture field for the same point in time is depicted in panel (b) of Figure 4.

The first simulation shows clear evidence of strong spatial imprint in the soil moisture fields that is easily identifiable because the shapes of the constituent river basins (Figure 4a) are apparent. Another interesting feature is a strong wet bias in a basin located in center of the Iberian Peninsula compared to its neighboring regions. Wet soils during this period are very unlikely because the entire region was enduring a prolonged and extreme drought. Moderate dry-bias is apparent in basins in southwest Germany, and a strong dry-bias was detected in basins in west Croatia, south Lithuania, south Hungary and north Bosnia and Herzegovina. Conversely, the soil moisture field obtained with the multi-basin parameter estimation does not exhibit these nuisances and thus can be regarded as a spatially seamless field. In this case, parameter estimation with a large sample of geophysical characteristics and many streamflow times series to estimate efficiency measures leads to a well-posed parameter estimation problem.

Based on these results, it can be concluded that parameter sets obtained using the on-site parameter estimation technique does not lead to seamless parameter fields or state variables. Moreover, automatic optimization algorithms, such as SCE or DDS, tend to over-learn from time series with large observational errors, which in turn leads to poor identifiability of param-

eters (Brynjarsdottir and O'Hagan, 2014) and biased simulations, as demonstrated above. Consequently, parameter estimation should be performed with a representative sample of basins that adequately cover the variability of hydrological regimes and geophysical properties (e.g., soil types) (Kumar et al., 2015). It is worth noting that if the parameters of a model are estimated in a small basin with very few soil types, a single geological formation, or very flat terrain, then it is very likely that some
parameters cannot be constrained during calibration. The obtained parameter set is biased to the specific basin in which it has been estimated and, hence, it is not skillful for seamless and continental scale simulations.

## 4.2    Effects of a parameterization technique on spatial patterns of effective parameters

The effects of the commonly used parameterization techniques to generate the porosity fields of LSMs (such as CHTESSEL and Noah-MP depicted in Figure 1) are important to investigate. These fields are obtained by combining the majority (or
dominant) upscaling operator and a look-up table containing categorical values of model parameters tabulated for a limited set of dominant soil types ( e.g., Niu (2011, p.20.), ECMWF (2016, p.137)). The majority-based operator is mostly used for estimating grid-specific vegetation classes in LSMs (Li et al., 2013).

The porosity field, based on a majority upscaling for the Noah-MP model used in EURO-CORDEX (www.euro-cordex.net) at an approximately 12 km resolution, is depicted in Figure 1. Compared with the other model derived porosity fields, the
Noah-MP field appears to be most homogeneously distributed in space. It is very likely that the spatial heterogeneity is under-represented in this case as the default soil LUT contains only thirteen soil classes. It should be noted that a model such as CABLE that uses a porosity field with an approximately 100 km resolution has a larger variability than that of Noah-MP at 12 km.

The following experiment is carried out to evaluate whether the variability of the soil map or the upscaling operator has a
larger effect on the derived porosity field. The highest resolution soil map available for Europe is used and applied in the same manner to derive porosity fields as described above. The texture field is provided by the SoilGrids dataset (soilgrids.org) at 1000 m resolution (level-0). The upscaled porosity field is generated at 5 km for the EDgE project. The soil characteristics for Noah-MP are estimated using the same look-up table as in the EURO-CORDEX-Noah-MP case. The comparison of both parameter fields (i.e., EDgE-Noah-MP and EURO-CORDEX-Noah-MP) and the main statistical moments describing the spatial
variability of the porosity fields are shown in Figure 5. The results clearly indicate the inappropriateness of the majority-based upscaling operator for this parameter in both cases. It leads to reduction of the variance of the porosity field and thus can be considered the least sensitive operator. This means that the informational content of the hyper-resolution soil maps, commonly available globally, is almost lost.

Notably, although the overall mean of the porosity estimated using MPR over the Pan-EU domain for mHM (Figure 3a) is
only 6.6% lower than that calculated using the majority-based approach for Noah-MP (Figure 5a), the spatial patterns obtained by both models are very different. The evidence of this remarkable dissimilarity can also be visualized by comparing the empirical density functions shown in Figures 3d and 5c, both corresponding to a field at $\ell_1 = 5$ km and with the same input data. A detailed evaluation conducted by Samaniego et al. (2012) in Germany showed that large porosity values estimated

with the majority-based approach could over-estimate those obtained with MPR by up to 40%, whereas in other locations, under-estimation up to 15% from those estimated by MPR can be found.

Other upscaling operators, such as the weighted arithmetic mean, are commonly used in LSMs in combination with the mosaic approach. For example, in CLM (Oleson et al., 2013, see p. 160) the texture class of the sub-units of the cell, called tiles, are provided in a look-up table. The upscaled porosity field obtained using this approach is shown in Figure 1 at a 1° (100 km) resolution. Methods based on the majority and weighted arithmetic mean operators exhibit some similarity and lack spatial variability. In both cases, the spatial mean is approximately 0.43 $m^3 m^{-3}$.

Hydrologic models that do not use soil porosity tend to use a similar conceptualization and values denoted as the total available water capacity (TAWC, WaterGAP versions 2 and 3) and field capacity (FC, HBV). For these types of conceptual models, normalized values of these parameters are used as surrogates for soil porosity. The consistency of the spatial patterns of TAWC and FC are compared here instead of their actual values. A distinctive difference in the patterns can be observed. For example, WaterGAP3 exhibits lower values than WaterGAP2, whereas the pattern of the normalized FC in HBV is the opposite in many locations (e.g., Spain, Germany, Scandinavia).

Details of the parameterization schemes used to estimate TAWC and FC are beyond the scope of this study. Interested readers may refer to Müller Schmied et al. (2014) or Beck et al. (2016), respectively. However, the TAWC in WaterGAP is obtained by linking the soil type provided by the FAO soil map with available water capacity values estimated by Batjes (1996). Thus, no scaling rule or form of regularization is used in this case. The field capacity parameters used in HBV were determined using an ad hoc nearest-neighbor interpolation technique that relies on calibrated parameters from nearby similar donor basins that might exhibit very different geophysical characteristics. The parameter fields obtained for two versions of WaterGAP (30 arcmin, 5 arcmin) and HBV are depicted in Figure 1. It can be concluded that the parameterization technique employed is not scale invariant as revealed by distinct parameter sets from WaterGAP model versions, which are operated at different resolutions. The regionalization proposed by Beck et al. (2016) leads to a patchwork-quilt field that does not resemble to any other field presented. Evident from the Figure Figure 1, the HBV field lacks seamlessness that may result in non-seamless fields of water fluxes and states.

## 4.3    Effects of a parameterization technique on the dynamics of a state variable

There is a complex interplay between soil moisture (SM) and latent heat (LH) in LSM/HMs. Improving our understanding of soil-land-atmosphere feedback is fundamental for making reliable predictions of water and energy fluxes. In this context, we carry out a sensitivity experiment to investigate the effects of soil related parameterizations (e.g., soil porosity) on latent heat and soil moisture. Two contrasting modeling paradigms (Noah-MP and mHM) are employed.

The WRF/Noah-MP system is forced with ERA-interim at the boundaries of the rotated CORDEX-Grid (www.meteo.unican.es/wiki/cordexwrf) at a spatial resolution of 0.11° covering Europe from 1989 to 2009. To ease the comparison, the process-based hydrological model mHM (www.ufz.de/mhm) is driven with daily precipitation and temperature fields generated by the WRF/Noah-MP system during the same period. The spatial resolution of mHM is fixed at 5×5 km². The main geophysical characteristics in WRF/Noah-MP of land cover and soil texture are represented with a 1×1 km² MODIS and a single-horizon,

coarse-resolution FAO soil map with 16 soil texture classes, respectively. The porosity field of Noah-MP is estimated by applying a majority-based operator to values for different soil classes, as shown in Figure 5b.

The settings of the mHM model used in this experiment are described in Section 3.4. In contrast with those of Noah-MP, the global parameters of mHM estimated using the MPR technique are obtained by closing the water balance over selected river basins in Europe (Rakovec et al., 2016a). The porosity fields obtained for mHM over the Pan-EU are depicted in Figure 3.

The phase diagrams of the monthly fraction of soil water saturation fSM $= \frac{\theta}{\theta_s}$ (i.e., plots of monthly fSM$(t)$ vs. fSM$(t+1)$) are subsequently investigated to understand the effect of differences in porosity estimates of the top 2 m soil column on the soil moisture dynamics (Figure 6). Two locations in Germany are selected in which Noah-MP systematically over- or underestimated the latent heat fluxes with respect to mHM (the latitude and longitude coordinates of the center of the selected Noah-MP grids are A: $(54°N, 10°E)$ and B: $(51°N, 7°E)$, respectively). At location A, the majority-based approach underestimates the MPR soil porosity by -10%, whereas in location B, it overestimates it by 40%. This experiment unambiguously shows that, at locations where Noah-MP over-estimates latent heat with respect to mHM, the temporal variance (i.e., dynamic) of the monthly SM time series simulated by Noah-MP is almost doubled compared to that of mHM, leading to much lower soil moisture values (Figure 6a). Conversely, underestimation of latent heat greatly reduces the variance of the soil moisture dynamics (Figure 6b).

## 4.4  Effects of not satisfying the flux matching condition

In Section 2, we postulated that ad hoc parameterization schemes do not necessarily fulfill the flux-matching test performed with a flux simulated by a given model at two modeling resolutions ( $\ell_1 = 5$ and 30 arcmin). A detailed description of how to perform this test is provided in Samaniego et al. (2010b). The following experiment is conducted with three models: mHM, PCR-GLOBWB, and WaterGAP in an attempt to falsify the above postulation. All models use the same forcings and geophysical information. The simulations are conducted in the Rhine River upstream of the Lobith gauging station. All three models are driven by daily forcing with a spatial resolution of 5 km, which was kindly provided by the EFAS team at JRC (www.eea.europa.eu). Additional details of the modeling settings of this experiment are provided in Sutanudjaja et al. (2015) and www.hyperhydro.org/. The KGE and bias values of these three models obtained for both scales at the Lobith station during 2003 are reported in Table 2. The daily streamflow time series during this year is selected for evaluation because it exhibits strong temporal dynamics, with wet conditions in the beginning of the year followed by a drought during the summer and fall seasons. The performances obtained for the three models are satisfactorily, but the results shown in Table 2 indicate that mHM is the only model that can have higher KGE values regardless of the spatial modelling resolution.

The flux-matching test presented in Section 3.1 is performed with simulated evapotranspiration (ET) because it is the largest flux in the water cycle besides precipitation, and is prone to the largest predictive uncertainties (Mueller et al., 2013). To ease the comparison, collocated grids are employed for every model such that every coarser scale grid cell has exactly the same number of underlying cells at finer resolution (5 arc min). The results of this test are shown in Figure 7a,b. They reveal that mHM exhibits the best flux-matching between these two scales. This experiment also shows that the MPR technique implemented in mHM leads to ET fields that are of similar magnitude at both scales indicating a close conservation of mass leading to the lowest relative errors (Figure 7c) among the three models.

The PCR-GLOBWB and WaterGAP models reveal large inconsistencies in preserving the spatial pattern of annual ET across two modeling scales, although the streamflow performance at the outlet is good (greater than 0.83 in both cases). PCR-GLOBWB at coarse resolution tends to understimate ET (up to 50%) compared with those at finer resolution (Figure 7f). Conversely, the coarser version of WaterGAP tends to overstimate ET (up to 60%) compared with those at the finer resolution (Figure 7i). Interestingly, it can be observed that changes in model resolution affect the dynamic of water fluxes in those models that do not use any consistent scaling rules for model parameterization. These results also confirm the postulation that "streamflow-related metrics are a necessary but not sufficient condition to warrant the proper partitioning of incoming precipitation P into various spatially distributed water storage components (e.g., SM) and fluxes (e.g., ET)" (Rakovec et al., 2016b). Because all models are forced with the same forcings, share the same geophysical information, and have almost similar hydrological process descriptions, it can be safely concluded that the parameterization method used in the models caused the ET mismatch. To falsify this postulation, the MPR parameterization protocol proposed in Section 3.3 is next applied to PCR-GLOBWB.

## 5   Implementation of the parameterization protocol in PCR-GLOBWB

To evaluate the consistency of land surface fluxes before and after MPR implementation, we analyze the impact of MPR on evaporative fluxes and soil moisture content in PCR-GLOBWB (van Beek et al., 2011; Wada and Bierkens, 2014; Sutanudjaja et al., 2016) over the Rhine River basin during 2003. The model is used to simulate the hydrological states at two different spatial resolutions ($\ell_1 =$5 and 30 arcmin), and the sensitivity to MPR implementation is evaluated using a field difference method (in line with eq. 1):

$$\Delta = \sqrt{\frac{1}{T} \sum_{t=1}^{T} \left(100 \frac{W_c(t) - w_f(t)}{w_f(t)}\right)^2} \tag{2}$$

where $W_c$ and $w_f$ are the coarse ($c$) and fine ($f$) resolution simulations of variable $W$, respectively, and $T$ is the total time series length.

The original PCR-GLOBWB parameterization does not include consistency in upscaling as enforced by MPR, leading to a larger difference in soil properties. Figure 8 depicts the porosity fields of this model before and after the implementation of MPR. Panels (a) and (b) of this figure show clearly the problems mentioned in section 2, for example lack of coherence in spatial patterns and the existence of spatial discontinuities of parameter fields at two scales. The porosity fields obtained with the MPR technique shown in panels (c) and (d), on the contrary, exhibit a typical seamless spatial structure in which the main features of the field can be distinguished across scales. It is worth noting that differences seen between Figure 8a and Figure 8c are not only due to the improved upscaling procedure, but also due to a modified pedo-transfer function. The parameters of the pedo-transfer function have also been included in the calibration within the MPR approach.

These differences in soil hydraulic properties influence the derived hydrological properties, leading to changes in saturated conductivity and storage capacity in the unsaturated zone. The considerable differences in ET fluxes are shown in panels (a)

and (b) of Figure 9 and are the result of these changes. When MPR is employed, we observe that the difference in actual average Rhine basin evapotranspiration between the two scales $\Delta$ drops from 29% to 9.4% (Figure 9d,e). For the total column soil moisture, we find a stronger decrease in $\Delta$ from 25% to 6.9%, clearly indicating the benefits of MPR implementation. The error fields in panels (c) and (f) of Figure 9 show a clear benefit of implementing MPR in PCR-GLOBWB. It should be noted, however, that the improvements are not as high as those obtained for mHM as shown in (Figure 7c). This is related with the fact that all effective parameters related with the evaporation and soil dynamic processes have been scaled with MPR in mHM, whereas in PCR-GLOBWB, only soil porosity has been scaled with this technique. Nevertheless, it is remarkable to see the improvements in flux-matching (Figure 9f) by scaling a single parameter of PCR-GLOBWB using MPR. We also observe a slight increase in the discharge performance (KGE) at Lobith. The original KGEs are 0.86 ($\ell_1$ =5 arcmin) and 0.93 ($\ell_1$ =30 arcmin), whereas the KGEs with MPR implementation are 0.91 and 0.93, respectively. Another advantage is that PCR-GLOBWB is calibrated at a coarser resolution, whereas this model is calibrated for each spatial resolution individually in the original set-up and with lower consistency in the discharge simulation.

From these evaluations, we conclude that MPR implementation leads to significant improvement in the flux matching and discharge simulations across scales, allowing for more consistency across scales for hydrological model simulations. Notably, additional parameters in PCR-GLOBWB still need to be regionalized within the MPR framework, which could potentially lead to better performance and transferability.

## 6 Conclusions

Hyper-resolution modeling initiatives (Wood et al., 2011; Bierkens et al., 2014) challenge the hydrological community to intensify efforts to make water (quantity and quality) and energy flux predictions "everywhere" and for these predictions to be "locally relevant." The predictions should have small uncertainties to be useful for the end-users. These grand challenges also imply that the next generation of land surface and hydrologic models must incorporate probabilistic descriptions of the sub-grid variability of geophysical land surface properties — such as POLARIS (Chaney et al., 2016b) and SoilGrids (Hengl et al., 2017) — to cope with the large uncertainties that characterize the related process below the Representative Elementary Area (REA) scale. Consequently, great efforts should be made in hyper-resolution monitoring at the global scale, in improving the computational efficiency of LSM/HMs, and in the development of scale-invariant parameterizations for these models. In this study, we have shown that the state-of-the-art parameterizations need to be improved to address this grand challenge, especially with respect to better fulfill the flux-matching condition.

We revisited a technique called Multiscale Parameter Regionalization (MPR) (Samaniego et al., 2010b), originally available only in mHM but recently implemented in PCR-GLOBWB as a part of this study. Moreover, we proposed a *Parameterization Protocol* as a guideline to apply MPR and to retrofit existing LSM/HMs to ease the implementation of MPR in the latter. We also discuss the advantages and limitations of MPR which should be considered while applying this concept to other LSM/HMs.

This study has shown that two models that use ad-hoc parameterizations can have reasonable efficiency with respect to simulated streamflow but poor performance with respect to distributed fluxes such as evapotranspiration. The implementation of this protocol in PCR-GLOBWB in this study increased the model efficiency by almost 6% and improved the consistency of simulated ET fields across scales. For example, the estimation of evapotranspiration without MPR at 5 arcmin and 30 arcmin spatial resolutions for the Rhine river basin resulted in a difference of approximately 29%. Applying MPR reduced this difference to 9%. For total soil water, the differences without and with MPR are 25% and 7%, respectively. We have also shown that the PCR-GLOBWB global parameters can be transferred across scales with consistent ET patterns and model efficiency.

In general, it can be concluded that the estimation of global parameters is feasible with MPR and that these scalars are transferable across scales and locations. The successful application of MPR implies that the averaging procedure of geophysical properties matters and that having the right physics with incorrect "effective" parameters leads to inconsistent fluxes and states. Consequently, MPR is a step forward to quasi scale-invariant parameterizations and is feasible to implement in existing LSM/HMs whose goal should be seamless parameter fields across scales that do not exhibits artificial spatial "discontinuities" such as calibration imprints, and that lead to consistent predictions across scales. We consider that this feature is the key for the next generation of LSM and NWP models such as the "model for prediction across scales" (MPAS) (www.mmm.ucar.edu) and the "nested-domain ICON" (www.earthsystemcog.org/projects/dcmip-2012/icon-mpi-dwd). Furthermore, a proper implementation of MPR in process based (conceptual) models may contribute to recent efforts towards identifying their "effective" parameters through observational datasets at the scale of interest (Savenije and Hrachowitz, 2017).

Finally, we would like to reiterate that a flux obtained from a land surface/hydrologic model should always be evaluated with local observations when available and across scales. If "it disagrees with experiment, it's wrong."

*Acknowledgements.* We kindly acknowledge our data providers: Noah-MP: Kirsten Warrach-Sagi (University of Hohenheim), PCR-GLOBWB: Niko Wanders (Princeton University), WaterGAP: Hannes Mueller-Schmied (University of Frankfurt), JULES: Anne Verhoef (The University of Reading), LISFLOOD: Peter Salamon (JRC), CABLE: Matthias Cuntz (formerly UFZ, now INRA), CLM: David Lawrence (UCAR) and E. Sutanudjaja and M. Bierkens, et al. for providing results from the HyperHydro WG1 Workshop 9-12 June 2015, Utrecht. This study was carried out within the Helmholtz-Association climate initiative REKLIM (www.reklim.de). This work has been partly funded by Helmholtz Alliance EDA - Remote Sensing and Earth System Dynamics, through the Initiative and Networking Fund of the Helmholtz Association, Germany. This study was partially performed under a contract for the Copernicus Climate Change Service (edge.climate.copernicus.eu). ECMWF implements this service and the Copernicus Atmosphere Monitoring Service on behalf of the European Commission. We thank Martin Schrön for kindly contributing to the artwork of Figure 2. Rens van Beek is acknowledged for providing support with the MPR implementation in PCR-GLOBWB. Finally, we would like to thank the Editor Murugesu Sivapalan and three anonymous reviewers for their insightful and helpful comments.

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

**Table 1.** Data sources and parameterization method used by models used in this study

| Model | Parameterization Method | References | Source code & Projects |
|---|---|---|---|
| CABLE | Pedo-transfer functions, look-up table, dominant soil type | Kowalczyk et al. (2006) | www.cawcr.gov.au/publications/technicalreports/CTR_057.pdf |
| CLM | Pedo-transfer functions, look-up table, mosaic approach | Oleson et al. (2013) | www.cesm.ucar.edu/models/cesm1.2/clm/ |
| CHTESSEL | Look-up table, dominant soil type | Viterbo and Beljaars (1995); ECMWF (2016) | www.ecmwf.int/search/elibrary |
| HBV | $k$-NN interpolation, calibrated parameter | Beck et al. (2016) | www.gloh2o.org/hbv-simreg/ |
| JULES | Look-up table, dominant soil type | Best et al. (2011) | jules.jchmr.org |
| LISFLOOD | Pedo-transfer functions, mosaic approach, arithmetic mean | De Roo and Wesseling (2000) | ec.europa.eu/jrc/en/publication/ eur-scientific-and-technical-research-reports/ lisflood-distributed-water-balance-and-/ flood-simulation-model-revised-user-manual-2013 |
| mHM | MPR | Samaniego et al. (2010b) | edge.climate.copernicus.eu www.ufz.de/mhm |
| Noah-MP | Look-up table, dominant soil type | Niu (2011) | www.jsg.utexas.edu/noah-mp www.meteo.unican.es/wiki/cordexwrf |
| PCR-GLOBWB | (Original) pedo-transfer functions with averaged predictors (New) MPR | van Beek et al. (2011); Wada and Bierkens (2014) Samaniego et al. (2010b) | pcraster.geo.uu.nl/projects/applications/pcrglobwb/ |
| WaterGAP (2,3) | Look-up tables | Müller Schmied et al. (2014); Batjes (1996) | www.uni-kassel.de/einrichtungen/en/cesr/research/ projects/active/watergap.html www.uni-frankfurt.de/45218063/WaterGAP |

**Table 2.** Efficiency of mHM, PCR-GLOBWB and WaterGAP obtained for the Rhine basin at Lobith station during 2003 for spatial resolutions of 5 and 30 arcmin.

| Model | 5 arcmin | | 30 arcmin | |
|---|---|---|---|---|
| | KGE | Bias [$m^3s^{-1}$] | KGE | Bias [$m^3s^{-1}$] |
| mHM | 0.96 | 61.19 | 0.96 | 21.74 |
| PCR-GLOBWB | 0.93 | -20.61 | 0.86 | 248.09 |
| WaterGAP (3,2) | 0.83 | 143.02 | 0.90 | -41.99 |

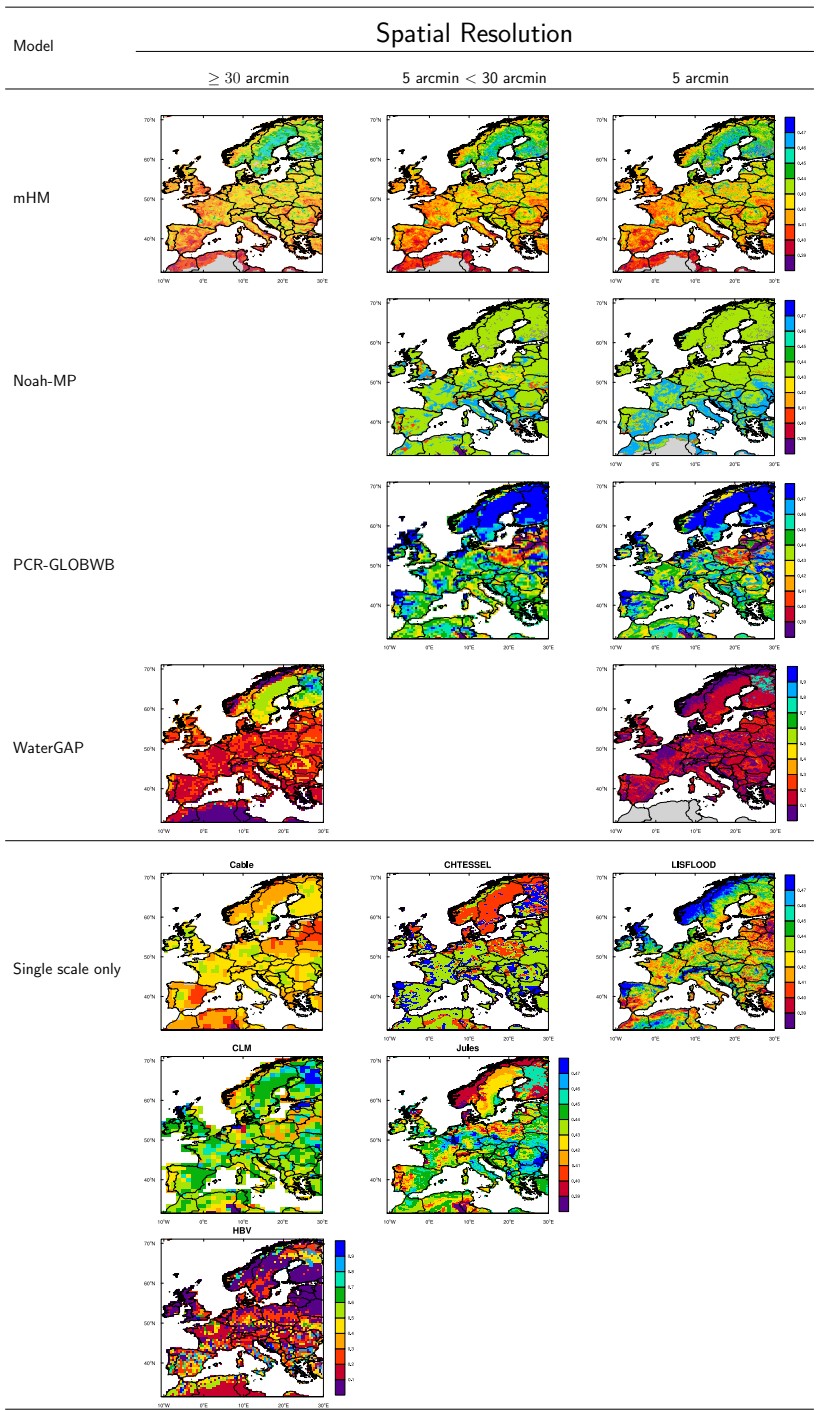

**Figure 1.** Porosity fields (top 2m) of typical LSM/HM over Pan-EU at various resolutions: CABLE (1°), CLM (1°), CHTESSEL (0.11°), JULES ( 35 km), LISFLOOD (EFAS, 5 km), mHM (EDgE-C3S, 5 km), Noah-MP (CORDEX-EU, 0.11°), and PCR-GLOBWB (EDgE-C3S, 5 km). Normalized available water capacity of WaterGAP2 (HyperHydro, 30 arcmin), [3,536] mm, WaterGAP3 (HyperHydro, 5 arcmin), [1,960] mm, and HBV [50,698] mm. The normalization values, denoted as [min,max], are provided only for HBV and WaterGAP.

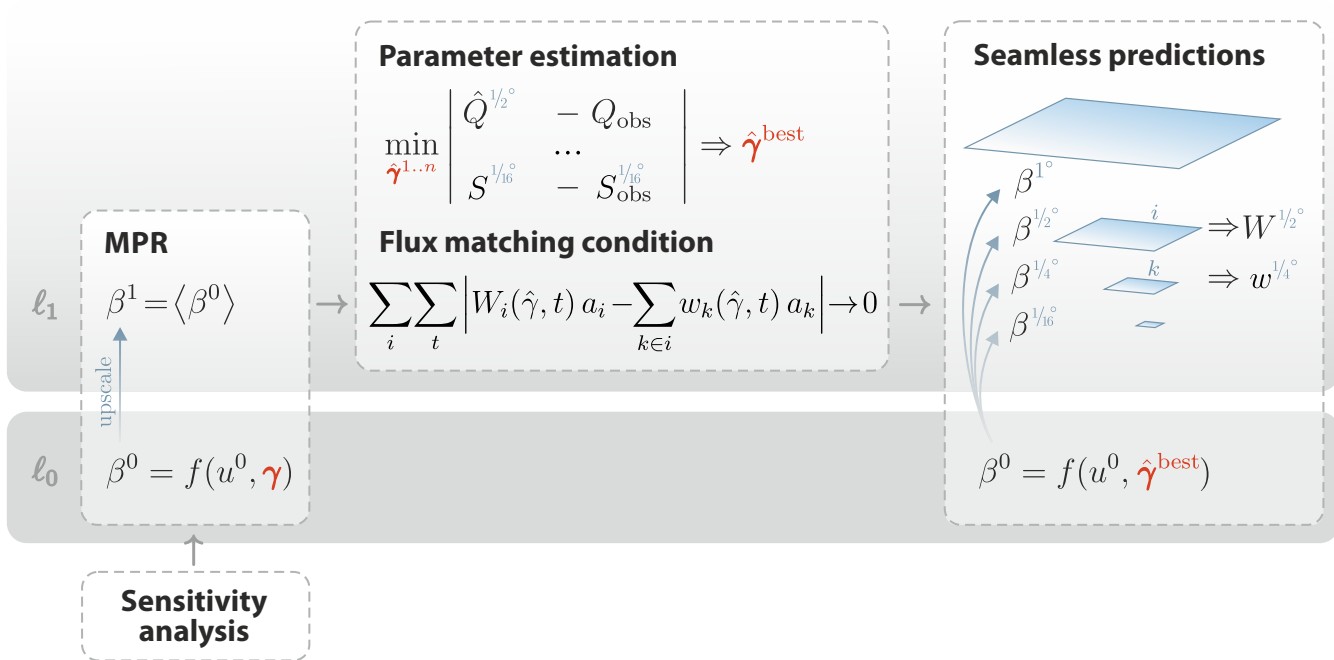

**Figure 2.** Schematic representation of the proposed seamless prediction framework based on Rakovec et al. (2016b). It includes a preliminary sensitivity analysis, MPR estimation, global-parameter estimation, a flux matching test, and multiscale seamless prediction. $W_i$ and $w_k$ are the fluxes at the $i$ and $k$ cells of the $1/2°$ and $1/4°$ resolutions, respectively (as an example). $Q_{obs}$ and $S_{obs}$ are the observed time series of streamflow and soil moisture, respectively. The operator $|\cdot|$ is a compromise dissimilarity metric composed of many independent observations at various scales.

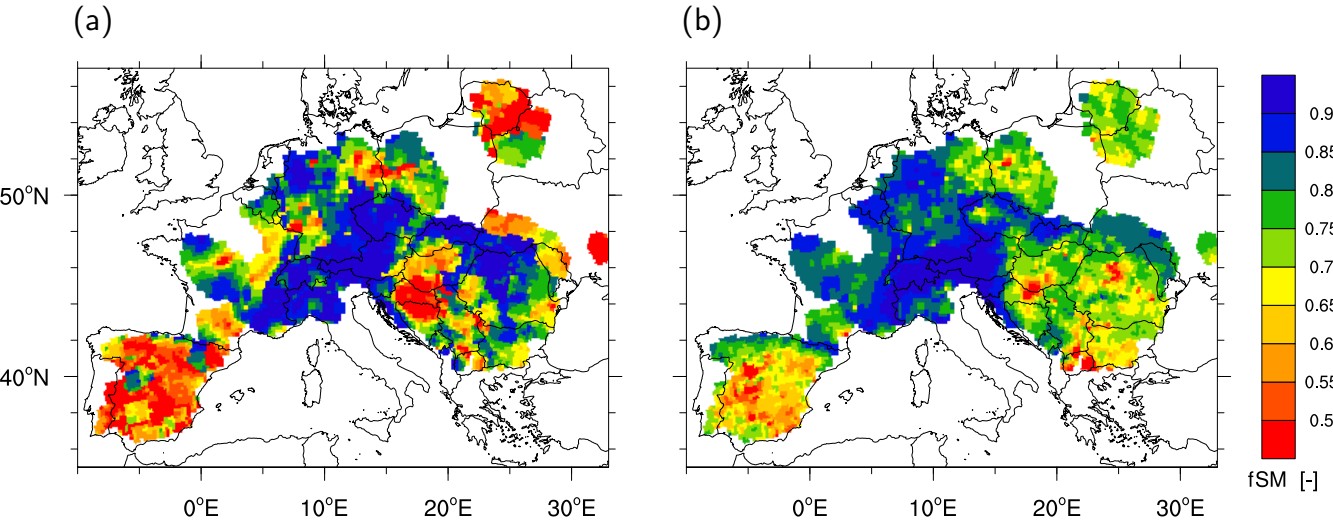

**Figure 3.** Seamless soil porosity (top 2 m) fields obtained using MPR at three spatial resolutions $\ell_1$ (a) 5 km, (b) 10 km, and (c) 25 km, respectively. Lower panels (d)-(f) show the empirical distribution function of porosity at the respective resolution and method.

**Figure 4.** mHM simulations of soil moisture as the fraction from saturation $\frac{\theta}{\theta_s}$ for a day in August/2005 conducted with (a) basin-wise parameter estimation and (b) seamless parameter estimation. Panel (b) shows a seamless soil moisture field.

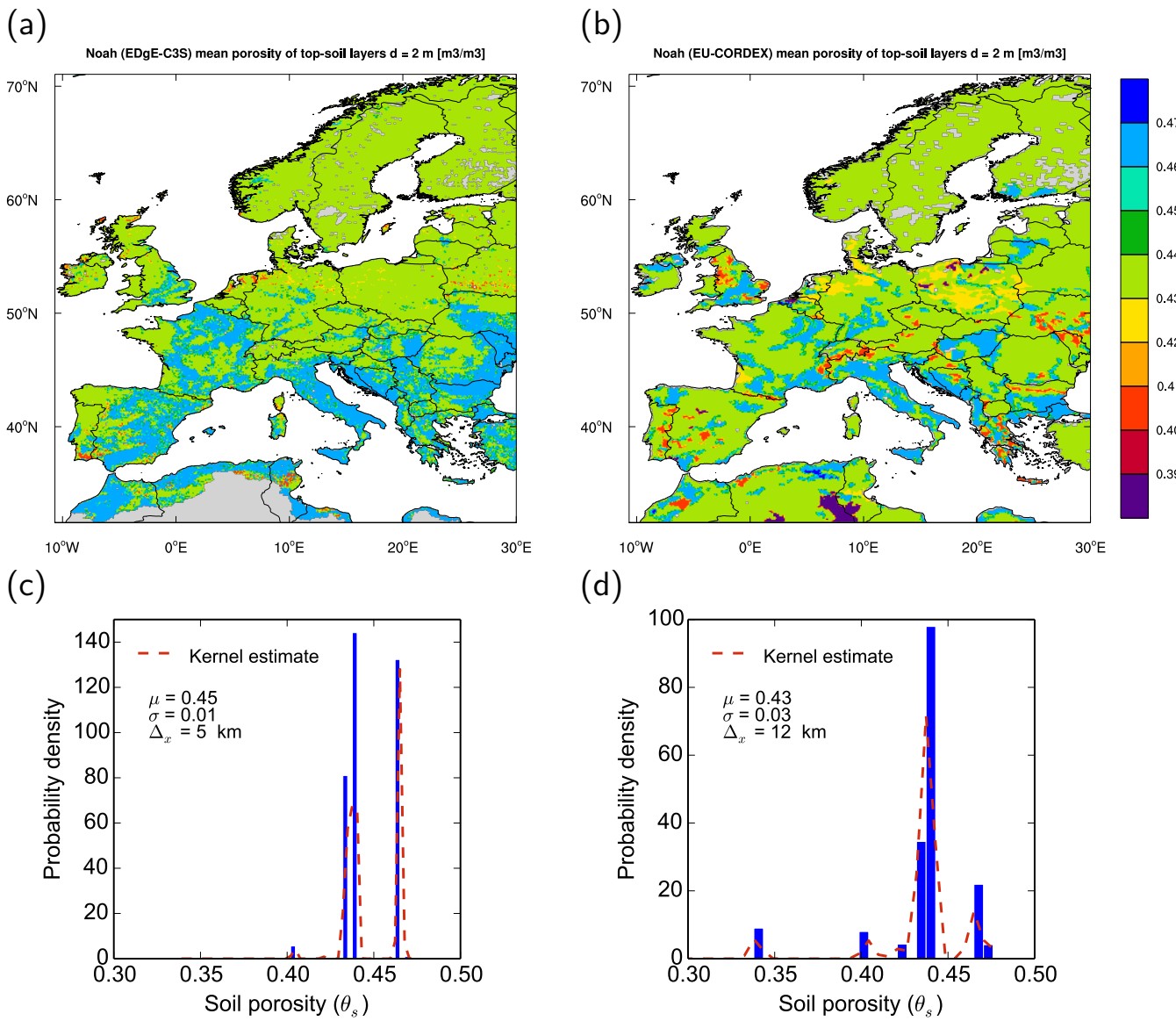

**Figure 5.** Porosity fields obtained using the majority upscale operator for spatial resolutions of (a) 5 km and (b) 12 km with the Noah-MP model used in the EDgE and EURO-CORDEX projects, respectively. Lower panels (c)-(d) show the empirical distribution function of porosity at the respective resolution and method.

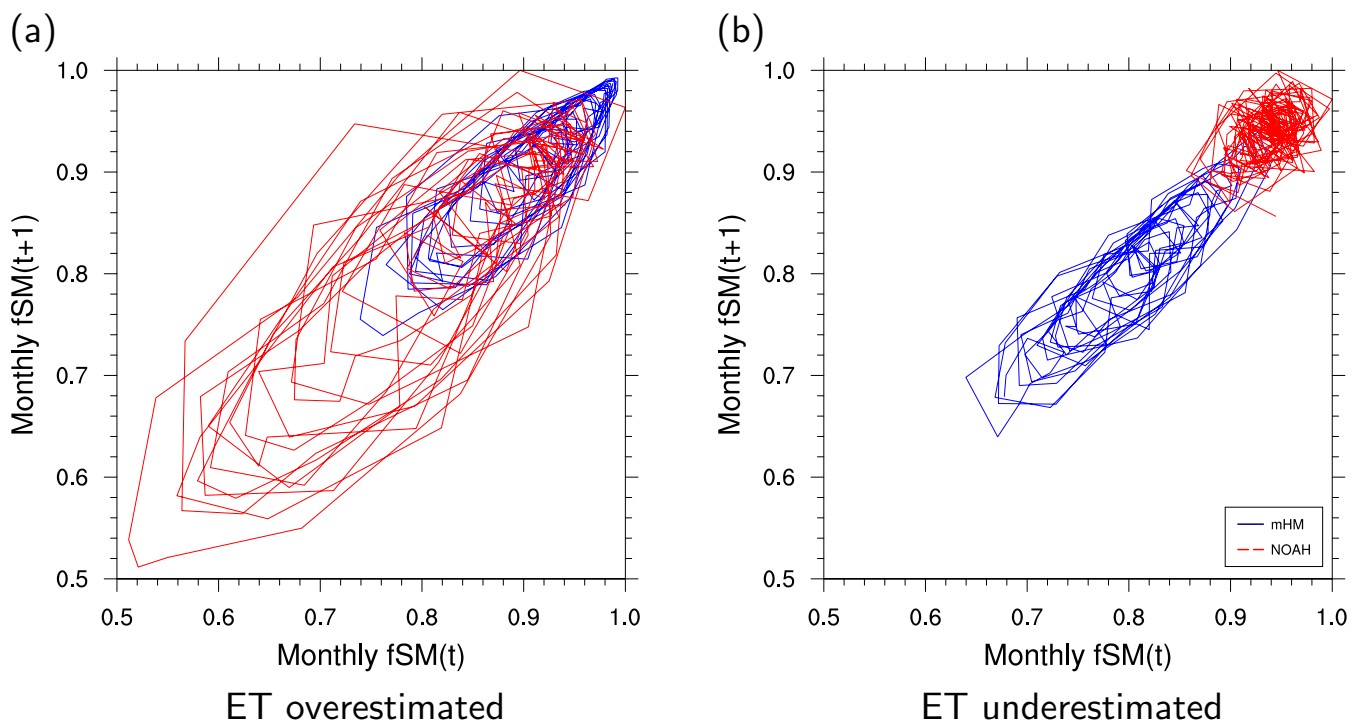

**Figure 6.** Phase diagrams of monthly soil moisture fraction for two locations in Germany, (a) $54°N,10°E$ and (b) $51°N,7°E$, in which the latent heat estimated by Noah-MP is over- or under-estimated with respect to corresponding estimates of mHM. The models have identical forcings.

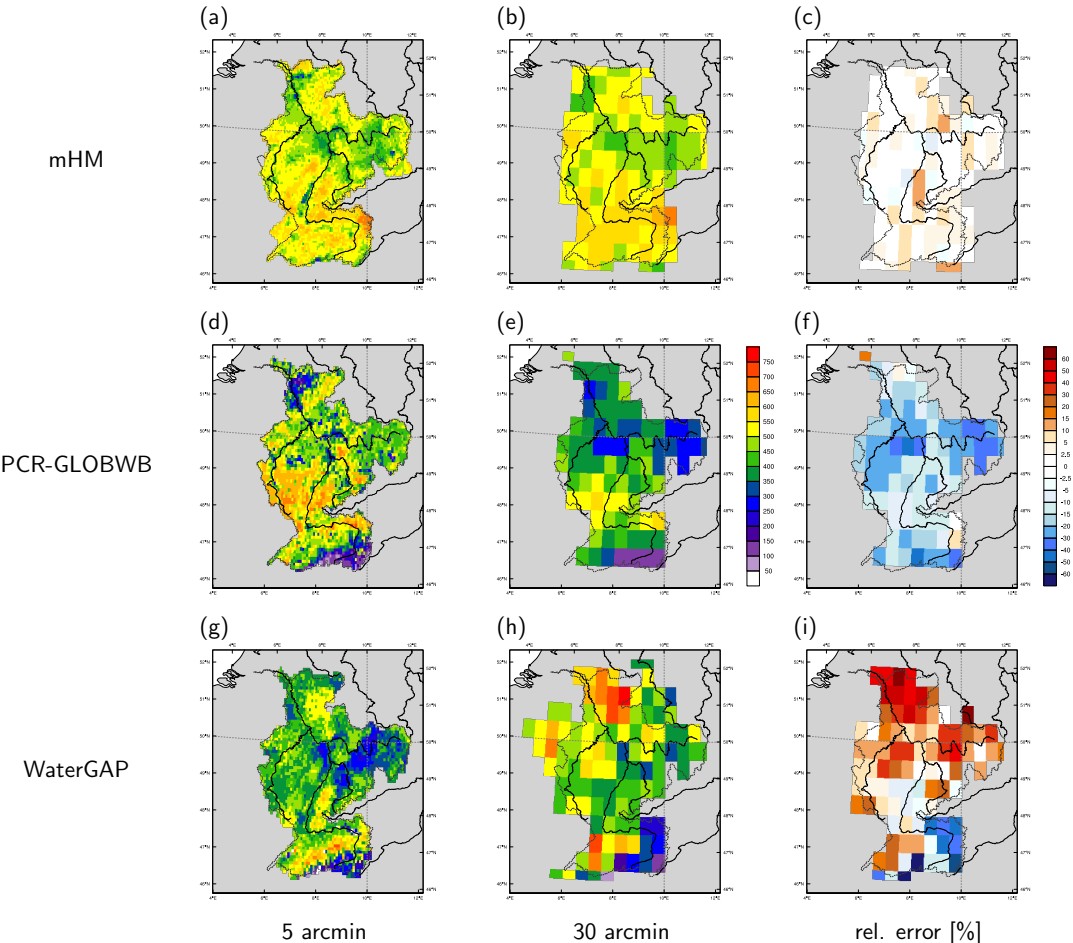

**Figure 7.** Multiscale simulation of annual ET for the Rhine River in 2003 with mHM, PCR-GLOBWB, and WaterGAP (versions 3 and 2) at spatial resolutions $\ell_1$ of 5 and 30 arcmin, respectively. The relative errors in percentage of the coarse field estimates with respect to the finer ones (aggregated to the coarser level) for mHM, PCR-GLOBWB, and WaterGAP are shown in panels (c), (f), and (i), respectively.

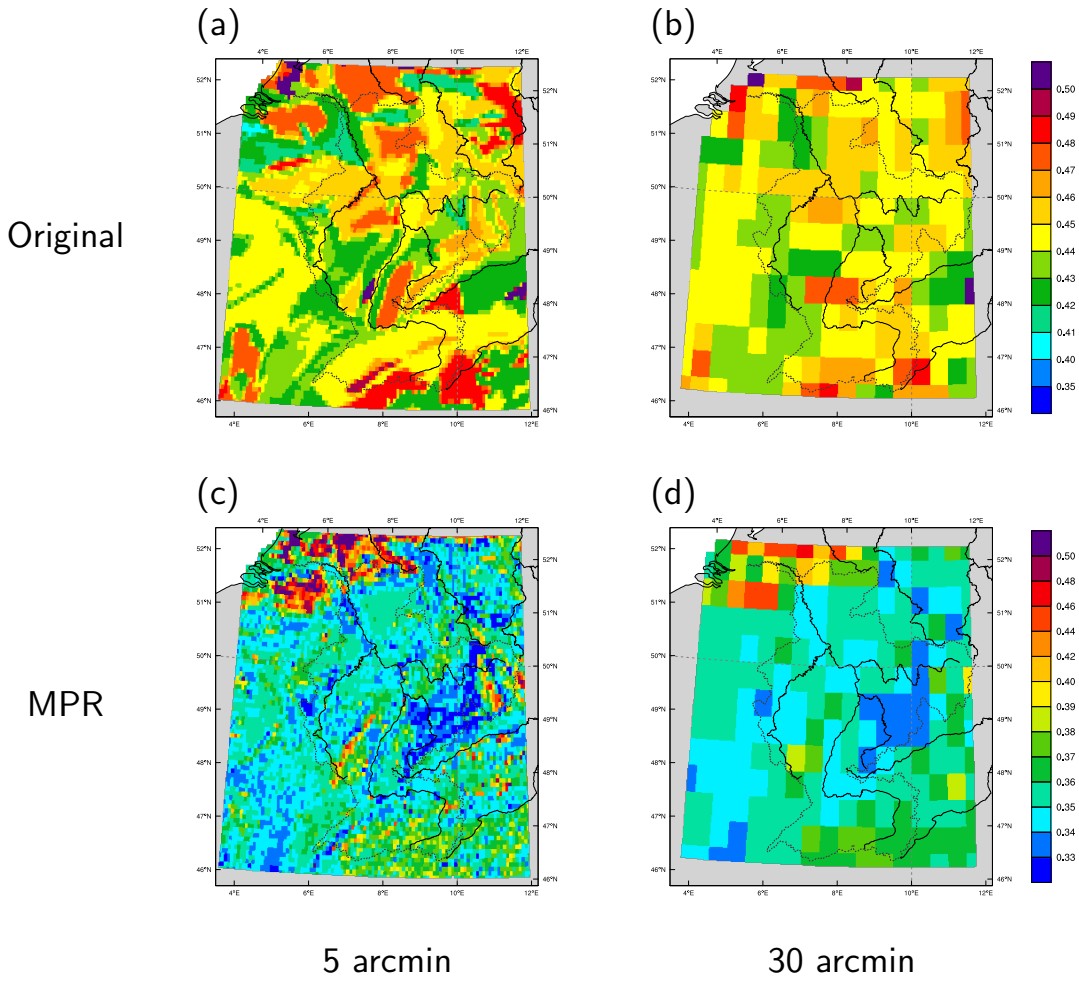

**Figure 8.** Porosity fields of PCR-GLOBWB before (panels a and b) and after implementing MPR (c and d) for two spatial resolutions of 5 and 30 arcmin. Dotted lines denote the Rhine basin and the continuous line is the main EU river basin network.

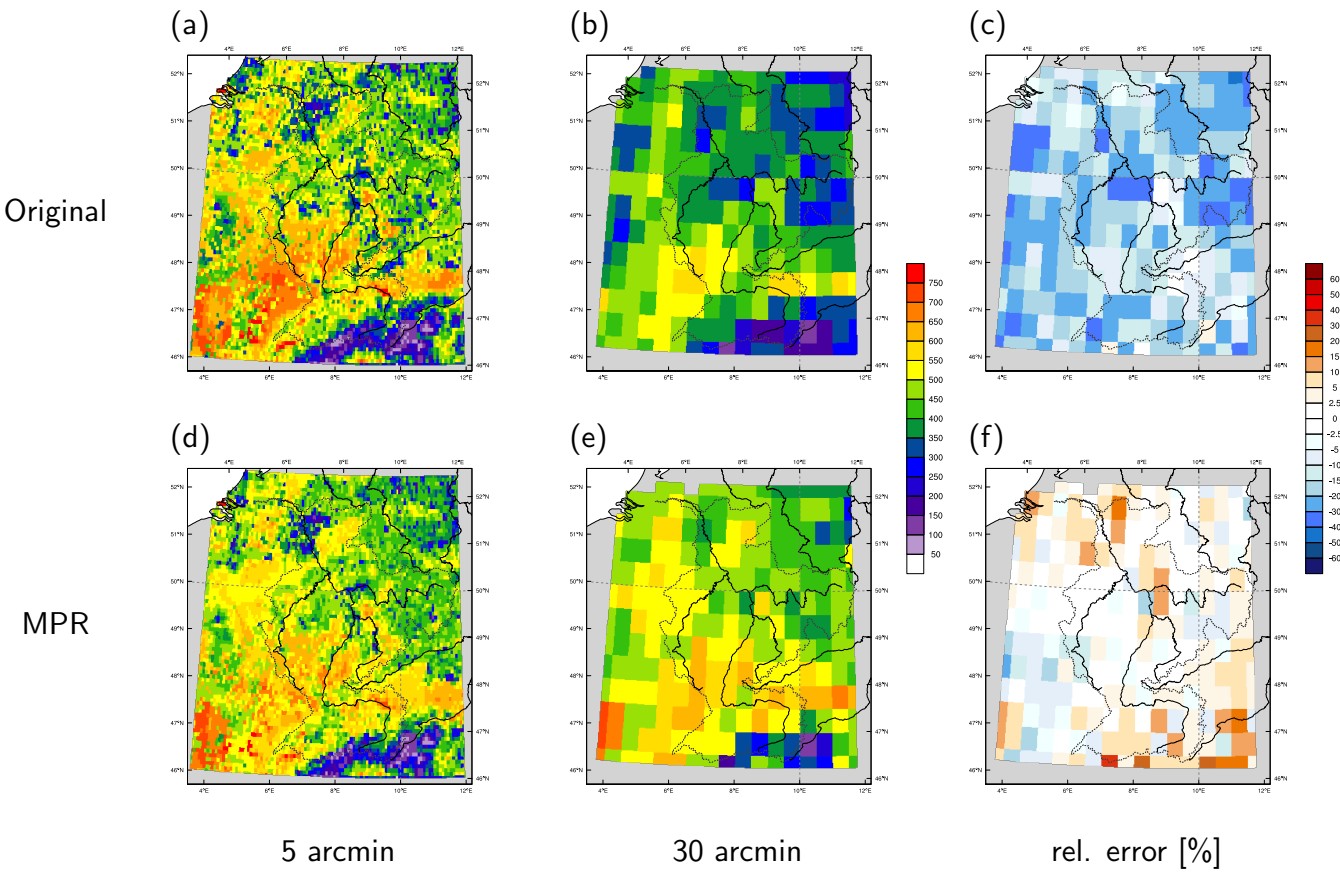

**Figure 9.** Annual ET fields in 2003 of PCR-GLOBWB before (panels a and b) and after implementing MPR (c and d) for two spatial resolutions of 5 and 30 arcmin. Dotted lines denote the Rhine basin and the continuous line is the main EU river basin network. The relative errors in percentage of the coarse field estimates with respect to the finer ones are shown in panels (c), (f) respectively.