# Peer review of "Toward seamless hydrologic predictions across spatial scales"

_Hydrology and Earth System Sciences, 2017_

## Referee Comment (RC1) · Anonymous Referee #1 · 17 Apr 2017

Samaniego et al. propose MPR to be a practical and robust method that provides consistent (seamless) parameter and flux fields across scales owing to the inconsistent and unrealistic parameter fields for land surface geophysical properties in many existing land surface and large-scale hydrological models. Although this study is properly motivated, I am having a hard time to understand what are the new advances from this manuscript comparing to Samaniego et al., WRR 2011 and Mizukami et al., 2017, particularly given that Mizukami et al. is submitted to WRR and perhaps under review.

Mizukami, N., Clark, M., Newman, A., Wood, A., Gutmann, E., Nijssen, B., Samaniego, L., and Rakovec, O.: Towards seamless large domain parameter estimation for hydrologic models, Water Resources Research, submitted., 2017

Another reason for my trouble of identifying new advances may be that lots of previous

concepts and methods (REA, REW, HRU etc.) are touched but in a rather scattered manner, i.e., without a coherent synthesis, thus making it difficult to follow the authors' logic chain to lead to the new contributions from this study. By briefly glancing through Samaniego et al., WRR 2011 I was guessing that perhaps in this study the major contribution is to introduce MPR as a robust parameter estimation approach for land surface and/or large-scale hydrological models, which in my mind are not really the same as those watershed-scale or highly-distributed hydrological models. For example, the application of MPR to PCR-GLOBWB has been largely illustrated in this manuscript. However, I am then confused again realizing there is another manuscript (Mizukami et al.) where MPR has also been applied to PCR-GLOBWB.

I therefore strongly encourage the authors clearly articulate the major advancements in this study. That said, I have a few specific comments as below.

1. L2, Page 2. "must made" –> "must be made" 2. L6, P10. It is not a good practice to jump from Fig. 2 to Fig. 7 (whilst Fig. 3-6 not introduced yet) 3. L6-8, P13. I don't think the argument so far can support this conclusion. Given the numerous processes controling the propagation from soil porosity to evapotranspiration and the fact these processes are very often presented & parameterized in different models with varying levels of complexity (i.e., model structure uncertainty), I could not really make sense out of this conclusion from my own experience (in both watershed modeling and land surface modeling) either. 4. L9-11, P13. As a modeler I could not agree with this conclusion either. A good parameter estimation method should never alter the true value of a parameter with very clear physical meaning, such as soil porosityA parameter, no matter at what resolution(s). Rather, the so-called predictive uncertainties mentioned here should be used a signature to diagnose whether the model itself is sufficiently robust, not the other way around. Otherwise, we are playing with the parameters to get the right anwer for the wrong reasons. 5. L26-27, P15. Why is this well-accepted fact (among modelers at least) being used as a hypothesis? 6. L10-11, P16. Don't follow the logic. According to L6-7, the majority-based approach in Noah-MP is giving 2.3%

HIGHER mean porosity than MPR. Why now the porosity field estimated by Noah-MP tends to have lower water holding capacity calues? 7. L19-21, P16. Does not read well. How could "dynamic(s)" be enhanced or constained? 8. L3-4, P17. Not so apparent to me. It appears to me PCR-GLOBWB does not perform bad either. But this may be due to the difficulty to link the flux-matching test with the spatial patterns here.
* * *

---

## Referee Comment (RC2) · Anonymous Referee #2 · 19 Apr 2017

The authors make a nice case for the value of their multiscale parameter regionalization (MPR) method, analysing several aspects (and advantages) of the method. This is in principle a laudable thing to do. The manuscript itself, however, is quite frustrating to read. One the one hand it remains completely unclear what the novelty is. Large parts of the manuscript essentially repeat what has already been published earlier (as also acknowledged in the references provided). On the other hand, the argument remains in places quite imprecise with a lot of quite sweeping (and not necessarily well substantiated) generalizations. In addition, other approaches to parameter selection are quite outrightly dismissed while essentially no critical discussion on potential drawbacks or limitations of MPR are provided. In an exaggerated way, the authors present their MPR method, which I think has formidable potential, like in a product promotion folder. I think the manuscript would strongly benefit from (1) considerably reducing the redundancies

with previous work (sections 1-3 can be *substantially* shortened) and (2) taking on a more critical perspective towards MPR. I think that many in the community will agree that it is a great tool. Instead of highlighting this over and over again, it would be more instructive to learn were its limitations are to allow further improvement.

In general, I think it may be more interesting for a wider audience if the MPR technique was scrutinized and compared to other parameter selection and regionalization approaches *independent* of the model it is used for. In this manuscript it is applied exclusively with mhm if I understand correctly. In my understanding, it is a stand-alone method that should be applicable to any model. Would it not be fairer to be more consistent in the comparisons here, i.e. compare mhm with/without mpr and/or other models with/without mpr?

The bottom-line is that I have the feeling that two quite independent things are not clearly separated here: the regionalization technique (MPR) and the models (mhm, etc). Here the text needs to become much more precise. Right now it seems to the reader that MPR is compared to e.g. the HBV model. This is not a valid comparison as these are completely different things. In contrast, it would be excellent to make the fact that MPR is a standalone tool clearer, as this may result in more modellers actually picking up the idea for their very own models (which they may not do at the moment due to its perceived exclusive association with mhm).

Specific comments:

p.2,l.5: why only over time and not also over space?

p.2,l.10-12: please avoid subjective terms as "elaborate" or "sophisticated"

p.2,l.28-29: is this actually true? Why would process dynamics that emerge at larger scales and that integrate several processes necessarily reduce "realism"? It is surely possible, but I do not think that it is a physical necessity. In any case, what is the meaning of "realism" in a situation where most of the system is de facto unobservable?

How do we know if something is "realistic"?

p.2,l.33-34: this is a sweeping generalization. What is actually meant by that? Why should an observed quantity, such as for example the stream flow recession constant have no physical meaning? Of course it has, albeit on the scale of the observation.

p.5,l.15ff and elsewhere: many things are mixed together here and the logic is not convincing. For a meaningful argument they need to be carefully disentangled. Is this about models? About parameter selection/calibration procedures? Parameter regionalization? It reads as if MPR does not rely on calibration, which is not correct. and why should lumped and/or semi-distributed models not be run with MPR-derived parameters? Would this for a, say 100km2 catchment, not be the same as if running a distributed model with a 10x10km2 grid in mhm?

p.5,l.19 and elsewhere in the manuscript: much is made of "discontinuities". However, the authors do not provide a clear definition of what they mean. Nature is, in places, discontinuous (e.g. forest vs. grassland, north vs. south aspect, sharp transitions in geology, breaks topography, etc). thus it is not clear why models should not represent these discontinuities. I suppose that the authors want to say that between individually calibrated catchments discontinuities can occur, where there are in reality no discontinuities. But this needs to be made clearer.

p.5,l.21-23: sure, but is this not also the case for distributed models and dependent on the calibration/parameter selection method?

p.6,l.29 and elsewhere: "CONUS": not necessarily every reader will be exposed to large scale studies employing these terms. Thus please avoid the use of fashionable abbreviations without first defining them.

p.8,l.7: a question cannot be postulated. Please rephrase.

p.8,l.10-11: what is meant by "poor". How do you define it?

p.9,l.3: over-parameterization is only addressed in MPR if simultaneously calibrated to

a high number of catchments and/or objective functions. Thus, it depends on how MPR is implemented and applied. Please rephrase.

p.10,l.17-18: how do you know that the parameters are "realistic"? See also comment above. Does this not also strongly depend on the assumptions in the upscaling relationships? It is always a question of how MPR (or other parameter selection techniques) are implemented and not a defining proprietary feature of MPR.

p.13, section 4: in many parts of the section it is unclear what is meant: the individual models or rather the parameter selection/regionalization techniques in the different model applications? These are different pairs of shoes and need to be carefully separated.

---

## Referee Comment (RC3) · Anonymous Referee #3 · 19 Apr 2017

**Toward seamless hydrologic predictions across scales**

**Samaneigo et al**

**Hydrol. Earth Syst. Sci. Discuss., doi:10.5194/hess-2017-89, 2017**

The main points of the paper are: (i) state-of-the-art LSMs and HMs do not have consistent and realistic parameter fields for land surface geophysical properties, and as a result do not satisfy a flux-matching condition (ii) the MPR technique can be used as a generic parameter estimation technique to greatly reduce these limitations (iii) a specific case of this improvement is demonstrated using the PCR-GLOBWB model.

In my view the innovation is in the recognition of the problem across multiple models, the wider breadth of application of MPR, and the protocol needed to achieve this. To some extent the purpose of the manuscript is to demonstrate the very significant consequences of different parameter estimation approaches in large-scale LSMs/HMs, and to show the advantages of using MPR. In my view this is a relevant objective for scientific publishing, in relation to relatively new techniques such as MPR, because such examples provide specific examples to which the hydrological modelling community can more easily relate (as opposed to reading about the MPR technique in the abstract, or in relation to its application to a specific model).

The main uncertainty for me is the extent to which this material is also contained in the submitted manuscript by Mizukami et al, as that manuscript is cited in relation to many of the main points made here. I leave this point for the Editor to consider.

1.  Title: "Toward seamless hydrologic predictions across scales" This might be interpreted by readers as referring to seamless predictions across temporal scales, i.e. the linking of nowcasting with NWP.  Perhaps "Toward seamless hydrologic predictions across spatial scales"?
2.  P2 L2 "trade-offs that must be made to reach a final objective" missing word
3.  P2 L9 "numerical weather prediction, land surface schemes, and hydrologic models" It would help to provide a reference or some text to enable readers to distinguish among these three terms. Many would know two of these terms, but far fewer could reliably distinguish all three.
4.  P2 L29 "In this case, one states that a physical process is parameterized." It would be helpful to introduce the concept of sub-grid phenomena here, to distinguish between phenomena which are resolved by a given grid resolution, and those that are parameterised. Otherwise, the concept of parameterisation and references to "the missing (complex) processes" remains rather vague. The missing processes should all be sub-grid – anything else that is missing is simply a missing process.
5.  P3 L1 "Parameterizations in land surface models have increased in their complexity during the past decades, but the procedures to estimate constants for the parameterizations have not changed much." Has anything changed as grid sizes got smaller? Did any processes become resolved that were formerly parameterized?
6.  P3 L7 "The reasons for the lack of progress in creating scale-invariant parameterizations are manifold." At this stage you have not established that scale-invariant parameterizations are either desirable or feasible (also relevant to P4 L24). From this point on in the paper it seems that the parameterization problem can be solved by scale-invariant parameterizations, but that there are no other credible paths being explored. I would like to see some mention in the Introduction of non-MPR approaches to parameter estimation which are also taking a

serious approach to the problem. Alternative methods are unlikely to satisfy the flux-matching criteria, but they might be partly competitive, e.g. (i) other spatial scaling attributes (e.g. sidestepping the scaling problem by assuming scale-independent distribution functions), (ii) strong links to mapped geophysical attributes (e.g. regularisation), (iii) strong links to observed functional responses of hydrological systems (e.g. Yadav et al (Advances in Water Resources 30 (2007) 1756–1774)).

7. P4 L19 "The numerical constants can be specified with a great level of precision, but the physical constants and parameters cannot be because they must be treated as random variables (Nearing et al., 2016)" I don't know the Nearing et al paper in detail, but I am surprised to hear that something termed a "physical constant" really requires treatment as a random variable. Surely if it is well enough defined to earn the moniker physical constant, then it can be determined experimentally to relatively high accuracy for practical purposes? Are the authors suggesting we should treat $g$ as a random variable in hydrology because it is determined by measurement, which is subject to error? On the other hand, I accept that parameters may usefully be described as random variables.

8. P4 I would like to see the term "seamless" defined in the introduction (the abstract provides this, but not the introduction), and particularly an argument made for why seamlessness is (in principle and/or in practice) a desirable attribute.

9. P9 The paragraph starting on L3 seems misplaced. The rest of the section is a description of MPR, whereas this paragraph is an assessment against criteria.

10. P9 L3 "Currently, MPR is the only method that consistently and simultaneously addresses the scale, nonlinearity and over-parameterization issues" If scale, nonlinearity and over-parameterization issues are the key criteria for assessment, then I would expect them to all be mentioned in the introduction; however, only scale really features in the introduction.

11. P9 L26 This whole paragraph (slightly rewritten) might sit well in the introduction if there was some material there on regularization procedures.

12. P9 L33 "Consequently, greater care should be taken in their selection." It is unclear what "greater" refers to. Are regularization functions being imposed without care? In which cases?

13. P11 L19 "Kling-Gupta efficiency (KGE) of the compromise solution > 0.6" Some justification is needed for any threshold on KGE, as it is much easier to do well in some environments than others.

14. P12 L3 "minimize the occurrence of discontinuities and ease the transferability of model parameters across scales and locations" These criteria for success should both have been outlined much earlier in the paper, either in the Introduction or at the end of the review.

15. P12 L17 "which constitute the basis for the EDgE project" Needs a reference to the project, or delete if not relevant.

16. P18 L28 "MPR … is feasible to implement in existing LSM/HMs whose goal should be seamless parameter fields across scales." The authors need to add an additional clause to this sentence (based on material from earlier in the paper) so it is clear WHY seamless parameter fields across scales are essential.

---

## Author Comment (AC1) · 1 May 2017

**Response to Referee #1**

1. *Samaniego et al. propose MPR to be a practical and robust method that provides consistent (seamless) parameter and flux fields across scales owing to the inconsistent and unrealistic parameter fields for land surface geophysical properties in many existing land surface and large-scale hydrological models. Although this study is properly motivated, I am having a hard time to understand what are the new advances from this manuscript comparing to Samaniego et al., WRR 2011 and Mizukami et al., 2017, particularly given that Mizukami et al. is submitted to*

*WRR and perhaps under review.*
*Mizukami, N., Clark, M., Newman, A., Wood, A., Gutmann, E., Nijssen, B., Samaniego, L., and Rakovec, O.: Towards seamless large domain parameter estimation for hydro- logic models, Water Resources Research, submitted., 2017*

Thank you for the comment. We are sorry for not making clear enough the differences between Mizukami et al. (under review) (hereafter [MCN+2017]) and this manuscript. We will make sure that this is clearly explained in the revised manuscript.
[MCN+2017] is aiming at the development of "a model agnostic MPR system called MPR-flex and then applied MPR-flex to the Variable Infiltration Capacity (VIC) model to produce hydrologic simulations over the contiguous USA (CONUS)". In [MCN+2017] no attempt has been made to verify the flux-matching condition of ET obtained with VIC using the MPR-flex parameterization across scales.

In this manuscript (hereafter [SKT+2017]) we attempt to describe the progress towards seamless parameterizations in land surface or hydrological models. We present a short description of what has been made (the literature on the topic is extensive) and provide a simple example to visualize how many of the existing models are estimating a fundamental parameter such as soil porosity. We postulate, based on our own experience, a way forward that uses MPR, provide a "Protocol for evaluation of model parameterization" (which is not publish before), implement it to PCR-GLOBWB (also new and unpublished) and carry out a series of experiments (based on the spirit of the E. Wood's recommendation) to demonstrate how to spot faulty parameterizations (also not publish before). We also compare the effects of the parameterization on three models (mHM, WaterGAP, and PCR-GLOBWB) as part of these experiments (all using the same forcings and underlaying data). It should be clearly noted that **any of these ba-**

**sic components are part of [MCN+2017].** In the revised manuscript, we will make clearer the scope of both papers.

2. *Another reason for my trouble of identifying new advances may be that lots of previous concepts and methods (REA, REW, HRU etc.) are touched but in a rather scattered manner, i.e., without a coherent synthesis, thus making it difficult to follow the authors' logic chain to lead to the new contributions from this study.*

   We will introduce a new table to synthesize the main developments related to concepts and methods (REA, REW, HRU etc.) used to obtain distributed parameter fields.

3. *... By briefly glancing through Samaniego et al., WRR 2011 I was guessing that perhaps in this study the major contribution is to introduce MPR as a robust parameter estimation approach for land surface and/or large-scale hydrological models, which in my mind are not really the same as those watershed-scale or highly-distributed hydrological models. For example, the application of MPR to PCR-GLOBWB has been largely illustrated in this manuscript. However, I am then confused again realizing there is another manuscript (Mizukami et al.) where MPR has also been applied to PCR-GLOBWB.*

   The hydrological process implemented in a land surface model (LSM) can be similar to those of a hydrological model(HM). We agree that LSMs and HMs are not the same because they aim at different purposes, and that the former ones tend to be much more complex than the later ones. The parameterization of soil parameters (e.g., soil porosity), however, can be based on the same principles of soil physics, and is often found in a large number of LSM/HM as shown in this manuscript.

   The reviewer's confusion may have been originated by weak formulations in p.9

   l.2 or in p.18 l.16. We will clarify these sentences in the revised manuscript. MPR has been applied to PCR-GLOBWB **only** in this manuscript up to now. The MPR-flex development presented in [MCN+2017] is applied **only** to VIC up to now.

4. *I therefore strongly encourage the authors clearly articulate the major advancements in this study. That said, I have a few specific comments as below.*

   Thank you for the recommendations. We will clarify them in the revised manuscript.

5. L2, Page 2. "must made" - "must be made"
   Done

6. L6, P10. It is not a good practice to jump from Fig. 2 to Fig. 7 (whilst Fig. 3-6 not introduced yet)
   Will be amended in the revised manuscript.

7. L6-8, P13. I don't think the argument so far can support this conclusion. Given the numerous processes controling the propagation from soil porosity to evapotranspiration and the fact these processes are very often presented & parameterized in different models with varying levels of complexity (i.e., model structure uncertainty), I could not really make sense out of this conclusion from my own experience (in both watershed modeling and land surface modeling) either.

   We are not claiming that MMS is better or worse than MPR. We are only comparing the values obtained by MMS w.r.t. those estimated by MPR and estimate the differences. For sure we do not now at this scale which values are more close to reality, the only fact we know is that the MPR estimates used in two HMs are good

enough to close the water balance in relatively well in over 300 basins over Pan-EU as shown in Rakovec et al. 2016. http://doi.org/10.1175/jhm-d-15-0054.1. We will clarify this formulation in the revised manuscript.

8. L9-11, P13. As a modeler I could not agree with this conclusion either. A good parameter estimation method should never alter the true value of a parameter with very clear physical meaning, such as soil porosity. A parameter, no matter at what resolution(s). Rather, the so-called predictive uncertainties mentioned here should be used a signature to diagnose whether the model itself is sufficiently robust, not the other way around. Otherwise, we are playing with the parameters to get the right anwer for the wrong reasons.

We do not agree with the reviewer in this point. An effective paramater at 5 km resolution or coarser is an "effective" parameter representing the heterogeneity of the underlying land surface and hence cannot be observed or measured directly but can only be estimated. Because of this fact, the effective values of porosity cannot compared directly with field samples. Binley, A., Elgy, J., & Beven, K. (1989). doi:10.1029/2008WR007695 and many other publications from Beven and Blöschl, Wood etc. make this fact very clear. If a model is applied at point scale (at most meters) then a parameter estimation method would lead to parameter vales that can be obtained in laboratory.

9. L26-27, P15. Why is this well-accepted fact (among modelers at least) being used as a hypothesis?

This "obvious fact" is used as a hypothesis to demonstrate that the Noah-MP model using the same soil map as mHM can lead to very different laten heat estimates although the forcings of both models are similar. We will reformulate the sentence and improve the text in the revised manuscript.

10. L10-11, P16. Don't follow the logic. According to L6-7, the majority-based approach in Noah-MP is giving 2.3% HIGHER mean porosity than MPR. Why

now the porosity field estimated by Noah-MP tends to have lower water holding capacity values?

Thank for pointing out this inconsistency. L6-7 refers to the mean over whole Pan-EU. L10-11 refers to a analysis in Germany whose results are reported in Fig. 6. We will clarify the text in the revised manuscript.

11. L19-21, P16. Does not read well. How could "dynamic(s)" be enhanced or constrained?

This text will be improved in the revised manuscript. We use the term "enhancing" to indicate changes in long-term mean and variance of soil moisture that occur by reducing/increasing the maximum water holding capacity (porosity times depth). We will make this point clear in the revised manuscript.

12. L3-4, P17. Not so apparent to me. It appears to me PCR-GLOBWB does not perform bad either. But this may be due to the difficulty to link the flux-matching test with the spatial patterns here.

If the parameters for both models are estimated based on streamflow only, then the model performance as reported in Table 2 tend to be comparable. ET estimates, however, differ greatly as shown in Fig.7 . In this case, both models in this experiment use collocated grids so that a cell at a coarser scale (30 arc min) have exactly the same number of underlying cells at finer resolutions (5 arc min), everywhere and for all models. Consequently, flux matching of ET made on two different resolutions is not a problem, and what is reported here is not an artifact of matching "spatial patterns".

---

## Author Comment (AC2) · 1 May 2017

The last sentence ofPoint 1 should read:
It should be clearly noted that **none of these basic components are part of [MCN+2017]**.
* * *

---

## Author Comment (AC3) · 1 May 2017

1. *The authors make a nice case for the value of their multiscale parameter regionalization (MPR) method, analysing several aspects (and advantages) of the method. This is in principle a laudable thing to do. The manuscript itself, however, is quite frustrating to read. One the one hand it remains completely unclear what the novelty is. Large parts of the manuscript essentially repeat what has already been published earlier (as also acknowledged in the references provided).*

We are saddened by the fact that the reviewer consider that this manuscript lacks novelty. We will improve the text to make it clear in the introduction. The novelty of the manuscript is based on the following key elements (see also the response

to Ref.1):

(a) Attempt to describe the progress towards seamless parameterizations in land surface(LSM) or hydrological models(HM). We present a short description of what has been made (the literature on the topic is quite extensive) and provide a simple example to visualize how existing LSMs/HMs are estimating a fundamental parameter such as soil porosity (not found in literature).

(b) We propose, based on our own experience, a way forward that uses MPR and systematize its application by providing a "Protocol for evaluation of model parameterization" (This has not been publish before)

(c) We implement this protocol to PCR-GLOBWB (also new piece of work and unpublished)

(d) Carry out a series of experiments (based on the spirit of the E. Wood's recommendation) to demonstrate how to spot faulty parameterizations (also not publish before).

(e) Compare the effects of the parameterization on three models (mHM, Water-GAP, and PCR-GLOBWB) as part of these experiments (all using the same forcings and underlaying data)

It should be clearly noted that **none of these key elements belong to Mizukami et al. (under review) (hereafter [MCN+2017])**.

2. *On the other hand, the argument remains in places quite imprecise with a lot of quite sweeping (and not necessarily well substantiated) generalizations.*

We will remove generalizations that are not fully substantiated with our experiments. It would be nice to know, however, which parts of our manuscript — according to the reviewer— are not substantiated enough.

3. *In addition, other approaches to parameter selection are quite outrightly dismissed while essentially no critical discussion on potential drawbacks or limitations of MPR are provided.*

We are not dismissing them. We simply do not have space in this study to test all of them. These evaluations (HRUs, Standard regionalization, etc.) have been carried out in independent studies which are cited in our manuscript. Here a short list:

(a) MPR vs. k-NN regionalization:
Samaniego, L., Bardossy, A., & Kumar, R. (2010). Streamflow prediction in ungauged catchments using copula-based dissimilarity measures. Water Resources Research, 46(2), http://doi.org/10.1029/2008WR007695

(b) MPR vs. standard regionalization (no scaling)
Samaniego, L., Kumar, R., & Attinger, S. (2010). Multiscale parameter regionalization of a grid-based hydrologic model at the mesoscale. Water Resources Research, 46(5), http://doi.org/10.1029/2008WR007327

(c) Lumped HRU, Distributed HRU, vs. MPR:
Kumar, R., Samaniego, L., & Attinger, S. (2010). The effects of spatial discretization and model parameterization on the prediction of extreme runoff characteristics. Journal of Hydrology, 392(1-2), 54-69. http://doi.org/10. 1016/j.jhydrol.2010.07.047

(d) MPR with satellite data (ungauged basin)
Samaniego, L., Kumar, R., & Jackisch, C. (2011). Predictions in a datasparse region using a regionalized grid-based hydrologic model driven by remotely sensed data. Hydrology Research, 42(5), 338-355. http://doi.org/10.2166/nh.2011.156

(e) MPR vs. HRU
Kumar, R., Samaniego, L., & Attinger, S. (2013). Implications of distributed hydrologic model parameterization on water fluxes at multiple scales and locations. Water Resources Research, 49(1), 360-379. http://doi.org/10.1029/2012WR012195

(f) MPR across scales US basins
Kumar, R., Livneh, B., & Samaniego, L. (2013). Toward computationally efficient large-scale hydrologic predictions with a multiscale regionalization scheme. Water Resources Research, 49(9), 5700-5714. http://doi.org/10.1002/wrcr.20431

(g) MPR in Pan-EU (transferability test, evaluation of states and fluxes more than 300 basins)
Rakovec, O., Kumar, R., Mai, J., Cuntz, M., Thober, S., Zink, M., et al. (2016). Multiscale and Multivariate Evaluation of Water Fluxes and States over European River Basins. Journal of Hydrometeorology, 17(1), 287-307. http://doi.org/10.1175/jhm-d-15-0054.1

Limitations and drawbacks of MPR w.r.t. to other methods have been mentioned in all our publications (see above). We will summarize them in the revised manuscript.

4. *In an exaggerated way, the authors present their MPR method, which I think has formidable potential, like in a product promotion folder.*

We politely disagree with the reviewer in this respect, basically for the following reasons. Our Manuscript cannot be consider a "promotion" folder because we put forward a protocol — NOT PUBLISHED before— and then proceed to apply it to

the model PCR-GLOBWB. Then we applied this model to two different scales to test the flux matching condition (all new). In addition to that we remark the deficiencies of current approaches to obtain seamless parameter fields (see fig. 1, also new).

5.  *I think the manuscript would strongly benefit from (1) considerably reducing the redundancies with previous work (sections 1-3 can be \*substantially\* shortened) and (2) taking on a more critical perspective towards MPR. I think that many in the community will agree that it is a great tool. Instead of highlighting this over and over again, it would be more instructive to learn were its limitations are to allow further improvement.*

    We will attempt to reduce redundancies. We recapitulate the MPR technique to have a self-consistent manuscript, if we move this section to an appendix, or refer MPR to other manuscripts, perhaps is not the optimal solution for the reader. We will try to summarize as much as possible though. As indicated above, limitations of MPR will be clearly written in the revised manuscript.

6.  *In general, I think it may be more interesting for a wider audience if the MPR technique was scrutinized and compared to other parameter selection and regionalization approaches \*independent\* of the model it is used for. In this manuscript it is applied exclusively with mhm if I understand correctly. In my understanding, it is a stand-alone method that should be applicable to any model. Would it not be fairer to be more consistent in the comparisons here, i.e. compare mhm with/without mpr and/or other models with/without mpr?*

    We politely disagree with the reviewer in this point for the following reasons. First of all, we can not attempt to repeat all was done in all the publications listed in point 3. Second, in this manuscript we are not only dealing with mHM but with PCR-GLOBWB and WaterGAP. Please see section 5! In this section **we fully**
**implement the protocol presented in this manuscript to PCR-GLOBWB**. This means with and without MPR in PCR-GLOBWB! Third, comparisons with/and without MPR in mHM have been done, please see Kumar et al 2010, Samaniego et al 2010ab, Kumar et al 2013, etc. and there will be soon a manuscript from Rakovec et al. over 500 US basins showing the effects of MRP/NO-MPR with mHM and VIC. Science is a cumulative enterprise from our point of view. We cannot repeat everything again and again.

7. *The bottom-line is that I have the feeling that two quite independent things are not clearly separated here: the regionalization technique (MPR) and the models (mhm, etc). Here the text needs to become much more precise. Right now it seems to the reader that MPR is compared to e.g. the HBV model. This is not a valid comparison as these are completely different things. In contrast, it would be excellent to make the fact that MPR is a standalone tool clearer, as this may result in more modellers actually picking up the idea for their very own models (which they may not do at the moment due to its perceived exclusive association with mhm).*

We politely disagree with the reviewer with some remarks in this point for the following reasons. First, we are comparing MPR with HBV! We are comparing parameters obtained with whatever method in various models that are related with the water holding capacity and porosity of the top soil and remarking that we have a problem with our LSMs/HMs that we need to be solved if we would like to have scale invariant parameterizations and consistent model simulations. MPR is a possible avenue, a hypothesis that we are scrutinizing over and over again in thousands of river basins across the globe. Second, mHM is an open source code available at www.ufz.de/mhm. Third, the model-agnostic version of MPR is called MPR-FLEX and is presented by [MCN+2017] and has been applied to VIC. Consequently, MPR is NOT exclusive from mHM now. We will improve the text so that this "impression" that "MPR is compared to e.g. the HBV model" vanish.

**Specific comments**

1. *p.2,l.5: why only over time and not also over space?*

   In this particular case, space is implicit and evolution refers to the development of spatial dependent variables over the time dimension. We will reformulate the sentence in the revised manuscript.

2. *p.2,l.10-12: please avoid subjective terms as "elaborate" or "sophisticated"*

   Done.

3. *p.2,l.28-29: is this actually true? Why would process dynamics that emerge at larger scales and that integrate several processes necessarily reduce "realism"? It is surely possible, but I do not think that it is a physical necessity. In any case, what is the meaning of "realism" in a situation where most of the system is de facto unobservable? How do we know if something is "realistic"?*

   We consider that this sentence is true. We do not want to start a philosophical discussion of what is "reality". We have a pragmatic approach, if a model is able to reproduce surrogate observations in evaluation mode, then we consider that the unobservable states may be plausible. For this reason, we carried out the study reported in Rakovec, et al. 2016 JHM (see above).

4. *p.2,l.33-34: this is a sweeping generalization. What is actually meant by that? Why should an observed quantity, such as for example the stream flow recession constant have no physical meaning? Of course it has, albeit on the scale of the observation.*

   We are referring here to transfer function parameters, for example those constants of the Clapp-Horberger PTF, which are basically found empirically and then used to link soil texture values (observable) with soil properties that may or

not be observable (e.g., porosity). We are not referring to streamflow recession constant. We will clarify the text to avoid confusions.

5. *p.5,l.15ff and elsewhere: many things are mixed together here and the logic is not convincing. For a meaningful argument they need to be carefully disentangled. Is this about models? About parameter selection/calibration procedures? Parameter region- alization? It reads as if MPR does not rely on calibration, which is not correct. and why should lumped and/or semi-distributed models not be run with MPR-derived pa- rameters? Would this for a, say 100km2 catchment, not be the same as if running a distributed model with a 10x10km2 grid in mhm?*

Based on this comment, we consider that the text is not correctly interpreted. We will clarify in the reviewed manuscript. Please refer to Samaniego et al. WRR 2010 to see a diagram that represent the steps done to estimate parameter for a given model. A simple conceptual model whose parameters are calibrated fail, in general, to perform well at cross-validation. This is what we are referring to. MPR improves transferability across scales and locations as shown in previous studies. In fact, this is what we demonstrate in Kumar et al. 2010 JoH. MPR could be used to estimate lumped parameters if a single cell covers the whole basin. In Kumar et al., mHM-MPR always performed better than a lumped mHM with no MPR.

6. *p.5,l.19 and elsewhere in the manuscript: much is made of "discontinuities". However, the authors do not provide a clear definition of what they mean. Nature is, in places, discontinuous (e.g. forest vs. grassland, north vs. south aspect, sharp transitions in geology, breaks topography, etc). thus it is not clear why models should not represent these discontinuities. I suppose that the authors want to say that between individually calibrated catchments discontinuities can occur, where there are in reality no discontinuities. But this needs to be made clearer.*

An example of artificially induced discontinuities by parameter calibration is

shown is Fig.4. We agree that there are natural discontinuities, we expect however, that it is unlikely that everywhere the model parameter and fluxes/state fields follow exactly the boundaries of the drainage area at a given location (see Fig.1 below). We call this negative effect calibration imprint, and we attempt to remove it with MPR. This artificial boundaries is what we call discontinuities. In the revised manuscript we will clarify our definition to avoid confusions. Nevertheless, we provide references to literature in p.5 l.19 to illustrate our definition. Please see the obtained parameter fields in Fig.1 (below) as obtained by Merz and Bloeschl 2004 and by MPR in Rakovec et al. 2016 JHM.

7. *p.5,l.21-23: sure, but is this not also the case for distributed models and dependent on the calibration/parameter selection method?*

   This is the case for any model even if one uses MPR on a single basin. This is the reason for showing the Fig.4a. Parameter estimation implies to have a representative sample. For this reason we attempt always to perform parameter estimation on several basins simultaneously, see Fig.4b. Single basin calibration is disadvantageous for any parameterization method because artifacts of the data can be "over-learned" which in-turn would induce large bias somewhere else.

8. *p.6,l.29 and elsewhere: "CONUS": not necessarily every reader will be exposed to large scale studies employing these terms. Thus please avoid the use of fashionable abbreviations without first defining them.*

   Done

9. *p.8,l.7: a question cannot be postulated. Please rephrase.*

   Thank you for the remark. We mean "put forward". It will be rephrased in the revised manuscript.

10. *p.8,l.10-11: what is meant by "poor". How do you define it?*

A poor parameterization does not lead to flux-matching, exhibits low model performances (say KGE) in cross-validation experiments across scales and locations, and exhibits artificial "discontinuities", i.e. non-seamless fields. This definition will be clearly mention in the revised manuscript.

11. *p.9,l.3: over-parameterization is only addressed in MPR if simultaneously calibrated to a high number of catchments and/or objective functions. Thus, it depends on how MPR is implemented and applied. Please rephrase.*

    We will rephrase this sentence in the revised manuscript.

12. *p.10,l.17-18: how do you know that the parameters are "realistic"? See also comment above. Does this not also strongly depend on the assumptions in the upscaling relationships? It is always a question of how MPR (or other parameter selection techniques) are implemented and not a defining proprietary feature of MPR.*

    This is a good question. It depends on many assumptions, PTFs, upscaling relationships, parameter estimation methods, etc. Visual impression may be useful but it is subjective. For these reasons, we need a formalized approach such as that described in Sec.3.3: **Protocol for evaluation of model parameterization**, which was put forward in this manuscript, and depicted in Fig.2. The experiments presented in Sec. 4 were introduced to addresses this question.

13. *p.13, section 4: in many parts of the section it is unclear what is meant: the individual models or rather the parameter selection/regionalization techniques in the different model applications? These are different pairs of shoes and need to be carefully separated.*

    We will clarify this section in the revised manuscript.
* * *
[Figure]

[Figure]

HRU based on: Merz & Bloschl, JoH 2004

[Figure]

MPR based on: Rakovec et al. JHM 2016

**Fig. 1.** Fields for the "Beta"parameter estimated for HBV and mHM. We consider that obtained with mHM and MPR a seamless field.

---

## Author Comment (AC4) · 1 May 2017

1. *The main points of the paper are: (i) state-of-the-art LSMs and HMs do not have consistent and realistic parameter fields for land surface geophysical properties, and as a result do not satisfy a flux- matching condition (ii) the MPR technique can be used as a generic parameter estimation technique to greatly reduce these limitations (iii) a specific case of this improvement is demonstrated using the PCR-GLOBWB model. In my view the innovation is in the recognition of the problem across multiple models, the wider breadth of application of MPR, and the protocol needed to achieve this. To some extent the purpose of the manuscript is to demonstrate the very significant consequences of different parameter estimation approaches in large-scale LSMs/HMs, and to show the advantages of using MPR. In my view this is a relevant objective for scientific publishing, in relation to*

*relatively new techniques such as MPR, because such examples provide specific examples to which the hydrological modelling community can more easily relate (as opposed to reading about the MPR technique in the abstract, or in relation to its application to a specific model). The main uncertainty for me is the extent to which this material is also contained in the submitted manuscript by Mizukami et al, as that manuscript is cited in relation to many of the main points made here. I leave this point for the Editor to consider.*

Thank you for the valuable comments and recommendations.

We described in detail the extend of Mizukami et al. (under review) (hereafter [MCN+2017]) and this manuscript in the Response to Referee #1 and Referee #2.

[MCN+2017] is aiming at the development of "a model agnostic MPR system called MPR-flex and then applied MPR-flex to the Variable Infiltration Capacity (VIC) model to produce hydrologic simulations over the contiguous USA (CONUS)". In [MCN+2017] no attempt has been made to verify the flux-matching condition of ET obtained with VIC using the MPR-flex parameterization across scales.

In this manuscript (hereafter [SKT+2017]) we:

(a) Attempt to describe the progress towards seamless parameterizations in land surface(LSM) or hydrological models(HM). We present a short description of what has been made (the literature on the topic is quite extensive) and provide a simple example to visualize how existing LSMs/HMs are estimating a fundamental parameter such as soil porosity (not found in literature),

(b) Propose, based on our own experience, a way forward that uses MPR and

systematizes its application by providing a "Protocol for evaluation of model parameterization" (This has not been publish before),

(c) Implement this protocol to PCR-GLOBWB (also new piece of work and un-published),

(d) Carry out a series of experiments (based on the spirit of the E. Wood's recommendation) to demonstrate how to spot faulty parameterizations (also not publish before), and

(e) Compare the effects of the parameterization on three models (mHM, Water-GAP, and PCR-GLOBWB) as part of these experiments (all using the same forcings and underlaying data)

It should be clearly noted that **none of these key elements belong to Mizukami et al. (under review) (hereafter [MCN+2017])**.

**Specific comments**

1. *Title: "Toward seamless hydrologic predictions across scales" This might be interpreted by readers as referring to seamless predictions across temporal scales, i.e. the linking of nowcasting with NWP. Perhaps "Toward seamless hydrologic predictions across spatial scales"?*

   Thank you for the good suggestion. Done.

2. *P2 L2 "trade-offs that must be made to reach a final objective" missing word*

   Done.

3. *P2 L9 "numerical weather prediction, land surface schemes, and hydrologic models" It would help to provide a reference or some text to enable readers to distinguish among these three terms. Many would know two of these terms, but far fewer could reliably distinguish all three.*

   References will be provided in the revised manuscript.

4. *P2 L29 "In this case, one states that a physical process is parameterized." It would be helpful to introduce the concept of sub-grid phenomena here, to distinguish between phenomena which are resolved by a given grid resolution, and those that are parameterised. Otherwise, the concept of parameterisation and references to "the missing (complex) processes" remains rather vague. The missing processes should all be sub-grid – anything else that is missing is simply a missing process.*

   Thank for the recommendation. The concept of sub-grid phenomena that are not modeled will be introduced in the the revised manuscript.

5. *5. P3 L1 "Parameterizations in land surface models have increased in their complexity during the past decades, but the procedures to estimate constants for the parameterizations have not changed much." Has anything changed as grid sizes got smaller? Did any processes become resolved that were formerly parameterized?*

   By comparing versions of land surface models, for example, multi-processes (parameterizations) have been introduced, e.g., in Noah-MP. Phenological processes and radiative transfer schemes have become extremely detailed in the new versions of Noah-MP and other LSMs. Runoff generation mechanisms, on the other hand, have not changed much in most LSMs/HMs. We will make a list of model improvements as grid sizes got smaller in the revised manuscript.

6. *P3 L7 "The reasons for the lack of progress in creating scale-invariant parameter-izations are manifold." At this stage you have not established that scale-invariant parameterizations are either desirable or feasible (also relevant to P4 L24). From this point on in the paper it seems that the parameterization problem can be solved by scale-invariant parameterizations, but that there are no other credible paths being explored. I would like to see some mention in the Introduction of non-MPR approaches to parameter estimation which are also taking a serious approach to the problem. Alternative methods are unlikely to satisfy the flux- matching criteria, but they might be partly competitive, e.g. (i) other spatial scaling attributes (e.g. sidestepping the scaling problem by assuming scale-independent distribution functions), (ii) strong links to mapped geophysical attributes (e.g. regularisation), (iii) strong links to observed functional responses of hydrological systems (e.g. Yadav et al (Advances in Water Resources 30 (2007) 1756–1774)).*

Good point. We will mention these alternative paths ways to be explored and we will discuss their main advantages and disadvantages in the revised manuscript. We consider, however, that it will be out the scope of this manuscript to test them.

7. *P4 L19 "The numerical constants can be specified with a great level of precision, but the physical constants and parameters cannot be because they must be treated as random variables (Nearing et al., 2016)" I don't know the Nearing et al paper in detail, but I am surprised to hear that something termed a "physical constant" really requires treatment as a random variable. Surely if it is well enough defined to earn the moniker physical constant, then it can be determined experimentally to relatively high accuracy for practical purposes? Are the authors suggesting we should treat g as a random variable in hydrology because it is determined by measurement, which is subject to error? On the other hand, I accept that parameters may usefully be described as random variables.*

Depends on the accuracy and precision with which we know a physical "constant". Its description can be done by a density function having a know mean and quite small standard deviation. For example, we know the value of the standard acceleration due to gravity with high accuracy (no bias) and precision(very small stdev). In this case and for practical purposes of parameter estimation, we could treat it as a constant. This is not necessarily the case for other physical constants such as the thermal conductivity of a given soil type. In this case with need a transfer-function of infer it based on soil texture fields and other predictors. We will clarify this statements in the revised manuscript.

8. *P4 I would like to see the term "seamless" defined in the introduction (the abstract provides this, but not the introduction), and particularly an argument made for why seamlessness is (in principle and/or in practice) a desirable attribute.*

   Good point. We will define it in the introduction of the revised manuscript to avoid miss interpretations. See definition in the Response to Referee #2, point 10.

9. *P9 The paragraph starting on L3 seems misplaced. The rest of the section is a description of MPR, whereas this paragraph is an assessment against criteria.*

   This paragraph will be relocated or rewritten in the revised manuscript.

10. *P9 L3 "Currently, MPR is the only method that consistently and simultaneously addresses the scale, nonlinearity and over-parameterization issues" If scale, nonlinearity and over- parameterization issues are the key criteria for assessment, then I would expect them to all be mentioned in the introduction; however, only scale really features in the introduction.*

    Good point. These issues were introduced in other publications related to MPR (e.g., Samaniego et al. 2010b). They will be introduced in the introduction of the revised manuscript.

11. *P9 L26 This whole paragraph (slightly rewritten) might sit well in the introduction if there was some material there on regularization procedures.*

We will use it in the introduction of the revised manuscript.

12. *P9 L33 "Consequently, greater care should be taken in their selection." It is unclear what "greater" refers to. Are regularization functions being imposed without care? In which cases?*

    If a regularization function is poorly chosen, or lack important predictors, the resulting parameter value might be badly estimated and its posterior distribution could be poorly estimated. For example, the Cosby et al. 1984 PTF is a very simple one (used in SCA-SMA) that relates porosity to sand content only. The application of this regularization function will under/over predict porosity in soils having low sand and high clay/loam fractions. We will mention this example to make clear our point in the revised manuscript.

13. *P11 L19 "Kling-Gupta efficiency (KGE) of the compromise solution > 0.6" Some justification is needed for any threshold on KGE, as it is much easier to do well in some environments than others.*

    This part of the protocol remains still subjective. It depends of on many factors such as the input forcings and quality of the land-surface properties. It is difficult to give a justification, but we will try to make it more objective recommendation in the revised manuscript.

14. *P12 L3 "minimize the occurrence of discontinuities and ease the transferability of model parameters across scales and locations" These criteria for success should both have been outlined much earlier in the paper, either in the Introduction or at the end of the review.*

    Good point. We will revise the introduction to mention them.

15. *P12 L17 "which constitute the basis for the EDgE project" Needs a reference to the project, or delete if not relevant.*

    We will add the reference to http://edge.climate.copernicus.eu

16. *P18 L28 "MPR ... is feasible to implement in existing LSM/HMs whose goal should be seamless parameter fields across scales." The authors need to add an additional clause to this sentence (based on material from earlier in the paper) so it is clear WHY seamless parameter fields across scales are essential.*

Good point. We will reformulate this section in the revised manuscript.

---

## Author Response (AR1)

**Response to Referee #1**

*Samaniego et al. propose MPR to be a practical and robust method that provides consistent (seamless) parameter and flux fields across scales owing to the inconsistent and unrealistic parameter fields for land surface geophysical properties in many existing land surface and large-scale hydrological models. Although this study is properly motivated, I am having a hard time to understand what are the new advances from this manuscript comparing to Samaniego et al., WRR 2011 and Mizukami et al., 2017, particularly given that Mizukami et al. is submitted to WRR and perhaps under review.*
*Mizukami, N., Clark, M., Newman, A., Wood, A., Gutmann, E., Nijssen, B., Samaniego, L., and Rakovec, O.: Towards seamless large domain parameter estimation for hydro- logic models, Water Resources Research, submitted., 2017*

Thank you for the comment. We are sorry for not making clear enough the differences between Mizukami et al. (under review) (hereafter [MCN+2017]) and this manuscript. In the revised manuscript we show clear differentiations between [MCN+2017] and our study, see P8 L7ff, P12 L16ff.
[MCN+2017] is aiming at the development of "a model agnostic MPR system called MPR-flex which is applied to the Variable Infiltration Capacity (VIC) model to produce hydrologic simulations over the contiguous USA (CONUS)". In [MCN+2017] no attempt has been made to verify the flux-matching condition of ET obtained with VIC using the MPR-flex parameter-ization across scales.

In this manuscript (hereafter [SKT+2017]) we attempt to describe the progress towards seam-less parameterizations in land surface or hydrological models. We present a short description of what has been made (the literature on the topic is extensive) and provide a simple example to visualize how many of the existing models are estimating a fundamental parameter such as soil porosity differently. We postulate, based on our own experience, a way forward that uses MPR, provide a "Protocol for evaluation of model parameterization" (which is not publish before), implement it to PCR-GLOBWB (also new and unpublished) and carry out a series of experiments (based on the spirit of the E. Wood's recommendation) to demonstrate how to spot faulty parameterizations (also not publish before). We also compare the effects of the parameterization on three models (mHM, WaterGAP, and PCR-GLOBWB) as part of these experiments (all using the same forcings and underlaying data). It should be clearly noted that none of these basic components are part of [MCN+2017].

*Another reason for my trouble of identifying new advances may be that lots of previous concepts and methods (REA, REW, HRU etc.) are touched but in a rather scattered manner, i.e., without a coherent synthesis, thus making it difficult to follow the authors' logic chain to lead to the new contributions from this study.*

These topics were excluded of the manuscript due to space restrictions and to improve the flow of the manuscript. We only refer to the HRU concept because it is commonly used for parameterization of HMs.

*... By briefly glancing through Samaniego et al., WRR 2011 I was guessing that perhaps in this study the major contribution is to introduce MPR as a robust parameter estimation approach for land surface and/or large-scale hydrological models, which in my mind are not really the same as those watershed-scale or highly-distributed hydrological models. For example, the ap-plication of MPR to PCR-GLOBWB has been largely illustrated in this manuscript. However, I am then confused again realizing there is another manuscript (Mizukami et al.) where MPR has also been applied to PCR-GLOBWB.*

The hydrological process implemented in a land surface model (LSM) can be similar to those of a hydrological model (HM). We agree that LSMs and HMs are not the same because they aim at different purposes, and that the former ones tend to be much more complex than the latter ones. The parameterization of soil parameters (e.g., soil porosity), however, can be based on the same principles of soil physics, and is often found in a large number of LSM/HM as shown in this manuscript.

The reviewer's confusion may have been originated by weak formulations in P9 L2 or in P18 L16 (old). We clarified these sentences in the revised manuscript. MPR has been applied to PCR-GLOBWB **only** in this manuscript up to now. The MPR-flex development presented in [MCN+2017] is applied **only** to VIC up to now. See P8 L7ff, P12 L16ff.

*I therefore strongly encourage the authors clearly articulate the major advancements in this study. That said, I have a few specific comments as below.*

Thank you for the recommendations. We explicitly point out the innovations of this study in the revised manuscript. The text of the introduction was drastically reduced to focus only on the state of the art that may lead to seamless parameterizations.

L2, Page 2. "must made" - "must be made"
Done

L6, P10. It is not a good practice to jump from Fig. 2 to Fig. 7 (whilst Fig. 3-6 not introduced yet)
Thank you for spotting this error. It was amended in the revised manuscript.

L6-8, P13. I don't think the argument so far can support this conclusion. Given the numerous processes controling the propagation from soil porosity to evapotranspiration and the fact these processes are very often presented & parameterized in different models with varying levels of complexity (i.e., model structure uncertainty), I could not really make sense out of this conclusion from my own experience (in both watershed modeling and land surface modeling) either.

We did not intent to claim that MMS is better or worse than MPR. We were only comparing the values obtained by MMS w.r.t. those estimated by MPR and estimate the differences. For sure we do not now at this scale which values are more close to reality, the only fact we know is that the MPR estimates used in two HMs are good enough to close the water balance in relatively well in over 300 basins over Pan-EU as shown in Rakovec et al. 2016. http://doi.org/10.1175/jhm-d-15-0054.1.
Since we do not really need the MMS data set in this study, and the comparison may lead to controversies, we decided to exclude these paragraphs from the revised manuscript.

L9-11, P13. As a modeler I could not agree with this conclusion either. A good parameter estimation method should never alter the true value of a parameter with very clear physical meaning, such as soil porosity. A parameter, no matter at what resolution(s). Rather, the so-called predictive uncertainties mentioned here should be used a signature to diagnose whether the model itself is sufficiently robust, not the other way around. Otherwise, we are playing with the parameters to get the right anwer for the wrong reasons.

Depends at the scale at which the PTF is applied. An effective paramater at 5 km resolution or coarser is an "effective" parameter representing the heterogeneity of the underlying land surface and hence cannot be observed or measured directly but can only be estimated. Because of this fact, the effective values of porosity cannot compared directly with field samples. Binley, A., Elgy, J., & Beven, K. (1989). doi:10.1029/2008WR007695 and many other publications from Beven and Blöschl, Wood etc. make this fact very clear. If a model is applied at point scale (at most meters) then a parameter estimation method would lead to parameter vales that can be obtained in laboratory. Since the MMS data set comparison was removed, this sentence does not appear in the revised manuscript.

L26-27, P15. Why is this well-accepted fact (among modelers at least) being used as a hypothesis?

Thank for mentioning it. Our intention in this experiment is a sensitivity analysis rather than a hypothesis testing. The text in the revised manuscript was revised accordingly.

L10-11, P16. Don't follow the logic. According to L6-7, the majority-based approach in Noah-MP is giving 2.3% HIGHER mean porosity than MPR. Why now the porosity field estimated by Noah-MP tends to have lower water holding capacity values?

Thank for pointing out this inconsistency. L6-7 refers to the mean over whole Pan-EU. L10-11 refers to a analysis in Germany whose results are reported in Fig. 6. The text in the revised manuscript was amended.

L19-21, P16. Does not read well. How could "dynamic(s)" be enhanced or constrained?

"Enhancing" and "constrained" are inappropriate terms. We should have written increasing or reducing the variance of soil moisture over time. This text was improved in the revised manuscript.

L3-4, P17. Not so apparent to me. It appears to me PCR-GLOBWB does not perform bad either. But this may be due to the difficulty to link the flux-matching test with the spatial patterns here.

If the parameters for both models are estimated based on streamflow only, then the model performance as reported in Table 2 tend to be comparable. ET estimates, however, differ greatly as shown in Fig.7 . In this case, both models in this experiment use collocated grids so that a cell at a coarser scale (30 arc min) have exactly the same number of underlying cells at finer resolutions (5 arc min), everywhere and for all models. Consequently, flux matching of ET made on two different resolutions is not a problem, and what is reported here is not an artifact of matching "spatial patterns". The text was improved to explain how the test was performed.

**Response to Referee #2**

*The authors make a nice case for the value of their multiscale parameter regionalization (MPR) method, analysing several aspects (and advantages) of the method. This is in principle a laudable thing to do. The manuscript itself, however, is quite frustrating to read. One the one hand it remains completely unclear what the novelty is. Large parts of the manuscript essentially repeat what has already been published earlier (as also acknowledged in the references provided).*

We are saddened by the fact that the reviewer consider that this manuscript lacks novelty. We have rewritten the introduction and modify large parts of the manuscript. The novelty of the manuscript is made more clear in the introduction and is based on the following key elements (see also the response to Ref.1):

1.1 Attempt to synthesize the progress towards seamless parameterizations in land surface(LSM) or hydrological models(HM). We provide examples to visualize how existing LSMs/HMs are estimating a fundamental parameter such as soil porosity (not found in literature) to make the case.

1.2 We propose, based on our own experience, a way forward that uses MPR and systematize its application by providing a "Protocol for evaluation of model parameterization" (This has not been publish before)

1.3 We implement this protocol to PCR-GLOBWB (also new piece of work and unpublished)

1.4 Carry out a series of experiments (inspired by E. Wood's recommendation) to demonstrate how to spot faulty parameterizations (also not published before).

1.5 Compare the effects of the parameterization on three models (mHM, WaterGAP, and PCR-GLOBWB) as part of these experiments (all using the same forcings and underlaying data). Also unpublished and novel material.

It should be clearly noted that none of these key elements belong to Mizukami et al. (under review) (hereafter [MCN+2017]). The main differences between [MCN+2017] and this manuscript can be found in P8 L7ff, P12 L16ff.

*On the other hand, the argument remains in places quite imprecise with a lot of quite sweeping (and not necessarily well substantiated) generalizations.*

We have removed generalizations that are not fully substantiated with our experiments. We would appreciate to know, which parts of our manuscript —according to the reviewer— need to be further substantiated, in the case that our justifications are not yet satisfactory.

*In addition, other approaches to parameter selection are quite outrightly dismissed while essentially no critical discussion on potential drawbacks or limitations of MPR are provided.*

There are a number of parameterizations approaches that have been tested in the literature. We provide a long list of references of comparisons between MPR and the most common techniques found in literature. These evaluations (HRUs, Standard regionalization, etc.) have been carried out in independent studies which are cited in our manuscript. Here a short list:

3.1 MPR vs. k-NN regionalization:
Samaniego, L., Bardossy, A., & Kumar, R. (2010). Streamflow prediction in ungauged catchments using copula-based dissimilarity measures. Water Resources Research, 46(2), http://doi.org/10.1029/2008WR007695

Limitations and drawbacks of MPR w.r.t. to other methods have been mentioned in all our publications (see above). Based on your recommendation, we provide in the revised manuscript a summary of the limitations of MPR (see new section 3.5).

*In an exaggerated way, the authors present their MPR method, which I think has formidable potential, like in a product promotion folder.*

We never intended that this manuscript is considered as a "promotion" folder because we conducted a series of new experiments, showed a novel application to PCR-GLOBWB and give recommendations regarding the MPR application for the scientific community. We make very clear in the introduction the aims and scope of the manuscript.

*I think the manuscript would strongly benefit from (1) considerably reducing the redundancies with previous work (sections 1-3 can be \*substantially\* shortened) and (2) taking on a more critical perspective towards MPR. I think that many in the community will agree that it is a great tool. Instead of highlighting this over and over again, it would be more instructive to learn were its limitations are to allow further improvement.*

We reduced redundancies and improved the introduction greatly. Introduction was shortened to focus on the main issue of the manuscript. We recapitulated the MPR technique to have a self-consistent manuscript, if we move this section to an appendix, or refer to MPR to other manuscripts, perhaps is not the optimal solution for the reader. We summarized as much as possible. The flux matching postulation has not been published in present form before. As indicated above, limitations of MPR will be clearly written in section 3.5 of the revised manuscript .

*In general, I think it may be more interesting for a wider audience if the MPR technique was scrutinized and compared to other parameter selection and regionalization approaches \*independent\* of the model it is used for. In this manuscript it is applied exclusively with mhm if I understand correctly. In my understanding, it is a stand-alone method that should be*

*applicable to any model. Would it not be fairer to be more consistent in the comparisons here, i.e. compare mhm with/without mpr and/or other models with/without mpr?*

We agree that the MPR method should be implemented to other models as well as model comparisons with and without MPR should be conducted. The first point is addressed within this manuscript by implementing MPR into PCR-GLOBWB using the herein proposed and developed protocol. Further implementations are underway such as MCN+2017 to VIC. We think the reviewer will admit that implementing a new parameterization technique like MPR goes along with a substantial adoption of model code which needs a lot of experience and knowledge of the model it is applied to. Therefore, several groups in the world are working on that a literature about such comparisons will increase. As you may admit it is already a substantial contribution to implement MPR to one model as shown herein (PCR-GLOBWB). Comparisons with/and without MPR in mHM have been done, please see Kumar et al 2010, Samaniego et al 2010ab, Kumar et al 2013, etc. and there will be soon a manuscript from Rakovec et al. over 500 US basins showing the effects of MPR/NO-MPR with mHM and VIC. Regarding the general comparison of parameterization techniques we consider a set of 11 different models (CABLE, CLM, CHTESSEL, JULES, LISFLOOD, mHM, Noah-MP, PCR-GLOBWB, WaterGAP2, WaterGAP3, and HBV) as a significant number of models.

*The bottom-line is that I have the feeling that two quite independent things are not clearly separated here: the regionalization technique (MPR) and the models (mhm, etc). Here the text needs to become much more precise. Right now it seems to the reader that MPR is compared to e.g. the HBV model. This is not a valid comparison as these are completely different things. In contrast, it would be excellent to make the fact that MPR is a standalone tool clearer, as this may result in more modellers actually picking up the idea for their very own models (which they may not do at the moment due to its perceived exclusive association with mhm).*

Within this study we never intended to compare MPR to other models such HBV. We adapted the text such that this possible confusions vanished. One of our intentions here is, however, to compare parameters obtained with different models and thus with different parameterizations. We have decided to take the water holding capacity and/or porosity of the top soil as an example since this parameter is used in all LSMs/HMs and is somehow physically interpretable. We are further remarking that we have a problem with our LSMs/HMs that we need to solve if we would like to have scale invariant parameterizations and consistent model simulations. MPR is a possible avenue, a hypothesis that we are scrutinizing over and over again in thousands of river basins across the globe. You are right that MPR is freely accessible. It comes with the mHM software package which is available at `www.ufz.de/mhm` as an open source code. Currently, a model-agnostic version of MPR (called MPR-FLEX) is developed and presented by [MCN+2017] and has been applied to VIC up to now. Consequently, MPR is NOT exclusive from mHM now. We have also improved the text so that this "impression" that "MPR is compared to e.g. the HBV model" has vanished.

**Specific comments**

*p.2,l.5: why only over time and not also over space?*

This paragraph is not appearing in the revised manuscript.

*p.2,l.10-12: please avoid subjective terms as "elaborate" or "sophisticated"*

Done.

*p.2,l.28-29: is this actually true? Why would process dynamics that emerge at larger scales and that integrate several processes necessarily reduce "realism"? It is surely possible, but I do not think that it is a physical necessity. In any case, what is the meaning of "realism" in a situation where most of the system is de facto unobservable? How do we know if something is "realistic"?*

This sentence was removed from the introduction. But, in this respect, we have a pragmatic approach, if a model is able to reproduce surrogate observations in evaluation mode, then we consider that the unobservable states may be plausible. For this reason, we carried out the study reported in Rakovec, et al. 2016 JHM (see above).

*p.2,l.33-34: this is a sweeping generalization. What is actually meant by that? Why should an observed quantity, such as for example the stream flow recession constant have no physical meaning? Of course it has, albeit on the scale of the observation.*

We are referring here to transfer function parameters, for example those constants of the Clapp-Horberger PTF, which are basically found empirically and then used to link soil texture values (observable) with soil properties that may or not be observable (e.g., porosity). We are not referring to streamflow recession constant. The word streamflow was removed to avoid confusions.

*p.5,l.15ff and elsewhere: many things are mixed together here and the logic is not convincing. For a meaningful argument they need to be carefully disentangled. Is this about models? About parameter selection/calibration procedures? Parameter region- alization? It reads as if MPR does not rely on calibration, which is not correct. and why should lumped and/or semi-distributed models not be run with MPR-derived pa- rameters? Would this for a, say 100km2 catchment, not be the same as if running a distributed model with a 10x10km2 grid in mhm?*

Based on this comment, we consider that our text is not clear enough. We reformulated the text to highlight the most common regionalization techniques used in recent literature. Please refer to Samaniego et al. WRR 2010 to see a diagram that represent the steps done to estimate parameter for a given model. A simple conceptual model whose parameters are calibrated fail, in general, to perform well at cross-validation. This is what we are referring to. MPR improves transferability across scales and locations as shown in previous studies. In fact, this is what we demonstrated in Kumar et al. 2010 JoH. MPR could be used to estimate lumped parameters if a single cell covers the whole basin. In Kumar et al., mHM-MPR always performed better than a lumped mHM with no MPR.

*p.5,l.19 and elsewhere in the manuscript: much is made of "discontinuities". However, the authors do not provide a clear definition of what they mean. Nature is, in places, discontinuous (e.g. forest vs. grassland, north vs. south aspect, sharp transitions in geology, breaks topogra- phy, etc). thus it is not clear why models should not represent these discontinuities. I suppose that the authors want to say that between individually calibrated catchments discontinuities can occur, where there are in reality no discontinuities. But this needs to be made clearer.*

Thank you for the comment. In the revised manuscript we clarify our definition to avoid confusions. An example of artificially induced discontinuities by parameter calibration is shown is Fig. 4. We agree that there are natural discontinuities, we expect however, that it is unlikely that everywhere the model parameter and fluxes/state fields follow exactly the boundaries of the drainage area at a given location (see Fig.1 below). We call this negative effect calibration imprint, and we attempt to remove it with MPR. This artificial boundaries is what we call discontinuities. Nevertheless, we provide references to literature in P4 L19ff to illustrate our definition. Please see also the obtained parameter fields in Fig.1 (rebuttal) (below) as obtained by Merz and Bloeschl 2004 and by MPR from the study Rakovec et al. 2016 JHM.

*p.5,l.21-23: sure, but is this not also the case for distributed models and dependent on the calibration/parameter selection method?*

This is the case for any model even if one uses MPR on a single basin. This is the reason for showing the Fig.4a. Parameter estimation implies to have a representative sample. For this reason we attempt always to perform parameter estimation on several basins simultaneously, see Fig.4b. Single basin calibration is disadvantageous for any parameterization method because artifacts of the data can be "over-learned" which, in-turn, induce large bias somewhere else.

[Figure]

HRU based on: Merz & Bloschl, JoH 2004

[Figure]

MPR based on: Rakovec et al. JHM 2016

Figure 1: Non-seamless vs. seamless parameter fields

*p.6,l.29 and elsewhere: "CONUS": not necessarily every reader will be exposed to large scale studies employing these terms. Thus please avoid the use of fashionable abbreviations without first defining them.*

Sorry for not defining it before as it should be. Done.

*p.8,l.7: a question cannot be postulated. Please rephrase.*

Thank you for this remark. We mean "put forward". It will be rephrased in the revised manuscript.

*p.8,l.10-11: what is meant by "poor". How do you define it?*

A poor parameterization does not lead to flux-matching, exhibits low model performances (say KGE) in cross-validation experiments across scales and locations, and exhibits artificial "discontinuities", i.e. non-seamless fields. This definition is be clearly mention in the revised manuscript (e.g., Introduction).

*p.9,l.3: over-parameterization is only addressed in MPR if simultaneously calibrated to a high number of catchments and/or objective functions. Thus, it depends on how MPR is implemented and applied. Please rephrase.*

This sentence is now amended as suggested in the revised manuscript.

*p.10,l.17-18: how do you know that the parameters are "realistic"? See also comment above. Does this not also strongly depend on the assumptions in the upscaling relationships? It is always a question of how MPR (or other parameter selection techniques) are implemented and not a defining proprietary feature of MPR.*

This is a good question. The word "realistic" was removed to avoid confusion. The text was amended to improve clarity. The application of MPR involve many assumptions, PTFs, upscaling relationships, parameter estimation methods, etc. Visual impression of parameter fields may be useful but it is subjective. For these reasons, we need a formalized approach such as that described in Sec. 3.3: **Protocol for evaluation of model parameterization**, which was put forward in this manuscript, and depicted in Fig.2. The experiments presented in Sec. 4 were introduced to addresses this question.

*p.13, section 4: in many parts of the section it is unclear what is meant: the individual models or rather the parameter selection/regionalization techniques in the different model applications? These are different pairs of shoes and need to be carefully separated.*

We renamed the experiments to help explain the intention of this section. We also added a short paragraph at the beginning of section 4 to elucidate the aim of these experiments. In summary, we performed these experiments to help identify poor parameterization techniques using several models.

**Response to Referee #3**

*The main points of the paper are: (i) state-of-the-art LSMs and HMs do not have consistent and realistic parameter fields for land surface geophysical properties, and as a result do not satisfy a flux- matching condition (ii) the MPR technique can be used as a generic parameter estimation technique to greatly reduce these limitations (iii) a specific case of this improvement is demonstrated using the PCR-GLOBWB model. In my view the innovation is in the recognition of the problem across multiple models, the wider breadth of application of MPR, and the protocol needed to achieve this. To some extent the purpose of the manuscript is to demonstrate the very significant consequences of different parameter estimation approaches in large-scale LSMs/HMs, and to show the advantages of using MPR. In my view this is a relevant objective for scientific publishing, in relation to relatively new techniques such as MPR, because such examples provide specific examples to which the hydrological modelling community can more easily relate (as opposed to reading about the MPR technique in the abstract, or in relation to its application to a specific model). The main uncertainty for me is the extent to which this material is also contained in the submitted manuscript by Mizukami et al, as that manuscript is cited in relation to many of the main points made here. I leave this point for the Editor to consider.*

Thank you for the valuable comments and recommendations.

We described in detail the extend of Mizukami et al. (under review) (hereafter [MCN+2017]) and this manuscript in the Response to Referee #1 and Referee #2. The main differences between [MCN+2017] and this manuscript can be found in P8 L7ff, P12 L16ff.

[MCN+2017] is aiming at the development of "a model agnostic MPR system called MPR-flex, which is applied to the Variable Infiltration Capacity (VIC) model to produce hydrologic simulations over the contiguous USA (CONUS)". In [MCN+2017] no attempt has been made to verify the flux-matching condition of ET obtained with VIC using the MPR-flex parameterization across scales.

In this manuscript (hereafter [SKT+2017]) we:

  1.1  Attempt to describe the progress towards seamless parameterizations in land surface(LSM) or hydrological models(HM). We present a short description of what has been made (the literature on the topic is quite extensive) and provide a simple example to visualize how existing LSMs/HMs are estimating a fundamental parameter such as soil porosity (not found in literature),

  1.2  Propose, based on our own experience, a way forward that uses MPR and systematizes its application by providing a "Protocol for evaluation of model parameterization" (This has not been publish before),

  1.3  Implement this protocol to PCR-GLOBWB (also new piece of work and unpublished),

  1.4  Carry out a series of experiments (based on the spirit of the E. Wood's recommendation) to demonstrate how to spot faulty parameterizations (also not publish before), and

  1.5  Compare the effects of the parameterization on three models (mHM, WaterGAP, and PCR-GLOBWB) as part of these experiments (all using the same forcings and underlaying data)

It should be clearly noted that none of these key elements belong to Mizukami et al. (under review) (hereafter [MCN+2017]) .

The scope of [MCN+2017] and [SKT+2017] are now clearly described in the first paragraph of Section 3.2 and in Section 3.5 (limitations of MPR). We also amended the conclusion that was misleading.

**Specific comments**

1. *Title: "Toward seamless hydrologic predictions across scales" This might be interpreted by readers as referring to seamless predictions across temporal scales, i.e. the linking of nowcasting with NWP. Perhaps "Toward seamless hydrologic predictions across spatial scales"?*

   Thank you for the good suggestion. Done.

2. *P2 L2 "trade-offs that must be made to reach a final objective" missing word*

   Done.

3. *P2 L9 "numerical weather prediction, land surface schemes, and hydrologic models" It would help to provide a reference or some text to enable readers to distinguish among these three terms. Many would know two of these terms, but far fewer could reliably distinguish all three.*

   Key references are provided in the revised manuscript.

4. *P2 L29 "In this case, one states that a physical process is parameterized." It would be helpful to introduce the concept of sub-grid phenomena here, to distinguish between phenomena which are resolved by a given grid resolution, and those that are parameterised. Otherwise, the concept of parameterisation and references to "the missing (complex) processes" remains rather vague. The missing processes should all be sub-grid – anything else that is missing is simply a missing process.*

   Thank for the recommendation. The concept of sub-grid phenomena that are not modeled will be introduced in the the revised manuscript.

5. *5. P3 L1 "Parameterizations in land surface models have increased in their complexity during the past decades, but the procedures to estimate constants for the parameterizations have not changed much." Has anything changed as grid sizes got smaller? Did any processes become resolved that were formerly parameterized?*

   By comparing versions of land surface models, for example, multi-processes (parameterizations) have been introduced, e.g., in Noah-MP. Phenological processes and radiative transfer schemes have become extremely detailed in the new versions of Noah-MP and other LSMs. Runoff generation mechanisms, on the other hand, have not changed much in most LSMs/HMs. We make this clear in the Introduction (P2, L20ff) of the revised manuscript.

6. *P3 L7 "The reasons for the lack of progress in creating scale-invariant parameterizations are manifold." At this stage you have not established that scale-invariant parameterizations are either desirable or feasible (also relevant to P4 L24). From this point on in the paper it seems that the parameterization problem can be solved by scale-invariant parameterizations, but that there are no other credible paths being explored. I would like to see some mention in the Introduction of non-MPR approaches to parameter estimation which are also taking a serious approach to the problem. Alternative methods are unlikely to satisfy the flux- matching criteria, but they might be partly competitive, e.g. (i) other spatial scaling attributes (e.g. sidestepping the scaling problem by assuming scale-independent distribution functions), (ii) strong links to mapped geophysical attributes (e.g. regularisation), (iii) strong links to observed functional responses of hydrological systems (e.g. Yadav et al (Advances in Water Resources 30 (2007) 1756–1774)).*

   Good point. Potential alternative ways are mentioned in the Introduction P3 L8ff. We consider, however, that exploring these alternative paths ways is out the scope of this manuscript to test them.

*P4 L19 "The numerical constants can be specified with a great level of precision, but the physical constants and parameters cannot be because they must be treated as random variables (Nearing et al., 2016)" I don't know the Nearing et al paper in detail, but I am surprised to hear that something termed a "physical constant" really requires treatment as a random variable. Surely if it is well enough defined to earn the moniker physical constant, then it can be determined experimentally to relatively high accuracy for practical purposes? Are the authors suggesting we should treat g as a random variable in hydrology because it is determined by measurement, which is subject to error? On the other hand, I accept that parameters may usefully be described as random variables.*

Depends on the accuracy and precision with which we know a physical "constant". Its description can be done by a density function having a know mean and quite small standard deviation. For example, we know the value of the standard acceleration due to gravity with high accuracy (no bias) and precision(very small stdev). In this case and for practical purposes of parameter estimation, we could treat it as a constant. This is not necessarily the case for other physical constants such as the thermal conductivity of a given soil type. In this case with need a transfer-function of infer it based on soil texture fields and other predictors. This section was, however, removed from the revised manuscript because the introduction was drastically cut to better focus on the topic of the manuscript.

*P4 I would like to see the term "seamless" defined in the introduction (the abstract provides this, but not the introduction), and particularly an argument made for why seamlessness is (in principle and/or in practice) a desirable attribute.*

Good point. We provided a definition in the introduction P2, L13 of the revised manuscript for consistency and to avoid miss interpretations.

*P9 The paragraph starting on L3 seems misplaced. The rest of the section is a description of MPR, whereas this paragraph is an assessment against criteria.*

This section was restructured in the revised manuscript to better explain the MPR approach.

*P9 L3 "Currently, MPR is the only method that consistently and simultaneously addresses the scale, nonlinearity and over-parameterization issues" If scale, nonlinearity and over- parameterization issues are the key criteria for assessment, then I would expect them to all be mentioned in the introduction; however, only scale really features in the introduction.*

Good point. These issues are briefly mentioned in the introduction. Many of these issues were introduced in other publications related to MPR (e.g., Samaniego et al. 2010b).

*P9 L26 This whole paragraph (slightly rewritten) might sit well in the introduction if there was some material there on regularization procedures.*

In this paragraph we analyze the effects of MPR on over-parameterzation. The Introduction was entirely rewritten, hence, we decided to keep this paragraph in this section. Thank you anyway for the suggestion.

*P9 L33 "Consequently, greater care should be taken in their selection." It is unclear what "greater" refers to. Are regularization functions being imposed without care? In which cases?*

If a regularization function is poorly chosen, or lack important predictors, the resulting parameter value might be badly estimated and its posterior distribution could be poorly estimated. For example, the Cosby et al. 1984 PTF is a very simple one (used in SCA-SMA) that relates porosity to sand content only. The application of this regularization function will under/over predict porosity in soils having low sand and high clay/loam fractions. See P2 L25, we mention this example to make clear our point in the revised manuscript.

*P11 L19 "Kling-Gupta efficiency (KGE) of the compromise solution ¿ 0.6" Some justification is needed for any threshold on KGE, as it is much easier to do well in some environments than others.*

This part of the protocol remains still subjective. It depends of on many factors such as the input forcings and quality of the land-surface properties. It is difficult to give a justification, it is only provided as a reference, and depends on the data quality, model, etc.

*P12 L3 "minimize the occurrence of discontinuities and ease the transferability of model parameters across scales and locations" These criteria for success should both have been outlined much earlier in the paper, either in the Introduction or at the end of the review.*

Good point. We introduced them in the introduction P2, L14.

*P12 L17 "which constitute the basis for the EDgE project" Needs a reference to the project, or delete if not relevant.*

We add the reference to `http://edge.climate.copernicus.eu` in the first reference to this project P11 L25.

*P18 L28 "MPR ... is feasible to implement in existing LSM/HMs whose goal should be seamless parameter fields across scales." The authors need to add an additional clause to this sentence (based on material from earlier in the paper) so it is clear WHY seamless parameter fields across scales are essential.*

Good point. We added a small clause to make the point. P19 L2

**Toward seamless hydrologic predictions across spatial scales**

[revised manuscript text omitted]

~~models) that exist today are the result of this elaborate formalization process, which led to a set of equations based on fundamental physical principles whose numerical solution is possible using sophisticated numeric algorithms and increasing computational resources. The scientific method used for modeling the water cycle has retained its fundamental structure since 1922, when L. Richardson wrote his seminal book in which the foundations for numerical weather forecasting were developed . A similar framework was employed by four decades later in formulation of the blueprint for a "distributed" hydrologic model.~~

~~Interestingly, "distributed" was introduced to distinguish this new type of model from the lumped, black-box hydrologic model that neglected the spatial variability of the input forcing data, state variables and fluxes, or the semi-distributed hydrologic models that partly accounted for this variability by subdividing the basin domain into sub-units (e. g., the Stanford Watershed Model) linked by river reaches.~~

An analysis of state-of-the-art LSMs and HMs reveals that most LSMs/HMs do not have consistent patterns of effective parameter fields for land surface geophysical properties across spatial scales, which indicates that their parameterizations are not scale-invariant. Parameter fields often exhibit artificial spatial "discontinuities" such as calibration imprints circumscribing river basins bound-aries, and consequently they are not seamless. There are several reasons explaining this parameterization deficiency. With the advent of electronic computers, the performance of general circulation models (GCMs), numerical weather pre-diction (NWP) models (Pielke Sr, 2013), land surface models (Liang et al., 1994; Sellers et al., 1997; Niu et al., 2011), and hydrologic models  (Batjes, 1996; Lindstrom et al., 1997; van Beek et al., 2011; Samaniego et al., 2010b) has been increased mainly by improving model conceptualization (i.e., the number of process descriptions) and/or spatial resolution  since the storage capacity and computational power allowed for it  .

~~Despite the above mentioned improvements in model development, "there are scales and physical processes that cannot be represented (or resolved) by a numerical model, regardless of the resolution" . Consequently, we simplify the process representations in our environmental models at the expense of physical realism. In this case, one states that a physical process is parameterized. A parameterization is a simplified and idealized representation of the physical phenomenon at a given scale. These simplifications require variables called predictors and constants, also called transfer-, global- or super-parameters which are often tuned to represent observed variables (e.g., streamflow); and have no physical meaning. These constants often constitute simplified surrogates to represent the missing (complex) processes that are not accounted for within a modeling system~~ .

 (Le Treut et al., 2007; Wood et al., 2011; Bierkens et al., 2014). As a result, parameterizations in LSMs have also increased in their complexity during the past decades  (Sellers et al., 1997; Fisher et al., 2014). The procedures to estimate  effective parameters required for the parameterizations~~have not changed much. It is possible to assert that model parameterization is an old, ubiquitous, and recurring problem in land surface and hydrologic modeling for which no final solution has been found. This lack of coherent development has induced to conclude that the "parameterization of hydrologic processes to the grid scale of general circulation models is a problem that has not been approached, let alone solved." A short survey of existing LSM/HMs presented in Section 2 allowed us to conclude that this statement is still valid~~, however, remained unchanged. For example, LSMs evolved from simple aerodynamic bulk transfer schemes with uniform description of surface parameters during the 1970s, to detailed LSMs having consistent description of the exchange of energy and matter between the atmosphere, the vegetation, and the land surface (Sellers et al., 1997). State-of-the-art LSMs, such as the Community Land Model version 4 (Bonan et al., 2011) and Noah-MP (Niu et al., 2011), however, still use quite simple pedotransfer-functions based on work of Clapp and Hornberger (1978) and Cosby et al. (1984) to estimate fundamental soil properties such as porosity (Oleson et al., 2013).

 Among the reasons that have prevented the improvement of parameterization techniques are: 1) the lack of procedures and theories for linking physical properties (e.g., soil porosity) that can be measured at the field scale with "effective" parameter values that represent the aggregate behavior of the land characteristics at the scale of a grid cell required in LSMs or HMs.

~~recognized that the theory and the "constants" required in the dynamic equations (hereafter called effective parameters) "must be appropriate to the size" of the grid element but suggested that these constants should be found experimentally (p.108), if possible. Decades later, suggested that parameters should be selected so that simulations could be extended to ungauged areas, and stated that, even with very detailed representative measurements, it will be "necessary to extrapolate results ...of physical parameters to other points of the basin." Linking effective parameters with point observations across a range of scales~~

~~implies a proper knowledge of scaling laws governing the phenomena at hand, the certainty of its invariance, and detailed knowledge of the spatial distribution of geologic formations and soil properties. Due to the non-linearity of the involved processes, extrapolation across scales, which are orders of magnitude apart, is very problematic. In this regard, concluded in his seminal paper that the state-of-the-art regarding "linking phenomena at field scales (10-100 ha) and catchment scales (10-1000 km$^2$) is an unresolved problem." There have been many attempts to bridge this gap, but the results have not been very~~

 2) Poor understanding of the scaling of parameters (Dooge, 1982) and its influence on the  hydrological response of the system (Wood, 1997; Wood et al., 1988). 3) Limited inclusion of sub-grid heterogeneity in hydrological parameterizations and multi-scale modeling of hydrologically relevant variables

~~In vadose zone hydrology, scaling attempts were pioneered by and followed by seminal works on fractal approaches , stochastic perturbation methods , stream tube approaches , and connectivity-based methods . These theories have allowed for finding relationships to scale hydraulic conductivity, pressure head, total porosity and other soil properties from the pore scale to the field scale and have shown that effective parameters may be scale-dependent . They have only been used to upscale evaporation and transpiration fluxes at the field scale and have not been used at larger scales until very recently. applied the~~

~~scaling proposed by to generate a global database of soil hydraulic properties. This data set, however, has not been used by any LSM/HMs up to now. The inverse modeling approach is frequently used to estimate soil-related parameters at regional scales . Stochastic and geostatistical theories have also been applied in saturated porous media for upscaling measured point-scale geophysical properties to the aquifer scale, including volume averaging theories and pre-asymptotic and asymptotic expansion theories. More recently, coarse graining methods were introduced to reduce the complexity of complex groundwater models~~

as suggested by Famiglietti and Wood (1995, 1994); Liang et al. (1996). 4) Lack of significant progress on the applicability of seminal upscaling theories (Miller and Miller, 1956; Dagan, 1989; Gelhar, 1993; Neuman, 2010; Kitanidis and Vomvoris, 2010) developed for sub-surface  hydrologic problems into LSM/HMs.

And 5) lack of transparency in most of the existing LSM/HM source  codes with respect to the meaning, origin and uncertainty associated with the hard-coded numerical values (i.e., parameters) either in the code or in the look-up tables
~~acceleration of gravity $g$ or $\frac{\pi}{2}$) and empirical effective parameters such as the soil porosity of a given soil type . noted that model output fluxes in the NOAH-MP model are sensitive to two-thirds of its applicable standard parameters, but most are hidden in the source code. From a statistical point of view, parameters and numerical constants are categorically very different. The numerical constants can be specified with a great level of precision, but the physical constants and parameters cannot be because they must be treated as random variables .~~(
[revised manuscript text omitted]

---

## Author Response (AR2)

**Response to Referee #1**

1 *I appreciate the efforts by the authors to address the comments from all reviewers. The writing, particularly the introduction part, has been largely improved. I now have a good appreciation of what's potentially novel in this study. The analysis, however, still does seem very clear to me, for example, overall I don't feel the current map presentation is straightforward to demonstrate the flux matching criteria hence the effectiveness of MPR in land surface modeling. I therefore strongly encourage the authors to further improve the presentation quality (now in the result section). I have some specific comments as below.*

Thank you for the kind words.

- *1. Page 13, L18, implement –¿ implementing*

  Done.

- *2. Page 22, L20, 22, Figure 9 is mentioned but I could not find Figure 9 from the previous and current version.*

  The new Fig. 9 was originally the Fig. 8 in the first version. The reference of the new Fig. 9 was provided in the 2nd version and can be found on page 36 of the previously submitted manuscript. Its caption starts as follows: "ET fields of PCR-GLOBWB before ...".

- *3. Figure 7 & 8, it'd be easier for readers to see the "seamless" effects by MPR if the authors could show the difference between ET simulations at different spatial resolutions. For example, perhaps the authors can simply aggregate the ET fluxes from Fig. 7a, c, e from 5 arcmin to 30 arcmin, then subtracted by the values in Fig. 7b, d, f..*

  The fields of relative bias, as suggested for Figs. 7 and 9, are now included in the revised figures and discussed in the manuscript. For Fig. 8, in our opinion, aggregation does not make sense because it refers to the porosity fields and not to water fluxes.

**Response to Referee #2**

1 *I am satisfied that the authors have satisfactorily addressed all the concerns I raised in my review.*
*We are happy to hear that.*
*However, the revised version has some minor issues in the Introduction that need addressing, as a result of it being restructured.*
*The introduction has 5 main points (in my opinion):*
*1. models run at different scales, but should produce consistent fluxes across scales*
*2. models have discontinuous parameter maps*
*3. the parameter discontinuities arise because complex models are difficult to parameterise; this remains a challenge for hydrology.*
*4. several methods have been proposed to address consistency of parameter fields, of which MPR is one that produces consistent fluxes across scales.*
*5. outline of the study*
*My main comments relate to Points 1 and 2 above:*

2 *Point 1. This seems reasonable but no justification is given here. Can you add half a sentence to say why flux-matching is so fundamental?*

This condition is key to ensure that a model satisfies the mass conservation principle at any scale. See Wood, E. (1997). Effects of soil moisture aggregation on surface evaporative fluxes (Vol. 190, pp. 397-412). Presented at the Journal of Hydrology. A sentence was added to make it clearer.

3 *Point 2. I was confused about whether parameter discontinuity was being presented as one of the results of this paper (in which case why do you have results in your introduction?), or as a result of another paper (in which case why are you not citing the previous work?).*

The sentence in P1 L11 ff in the previous version was removed to avoid confusion. The new sentence points out to "discontinuities" found in existing literature which are cited now. In this study we conduct a survey to demonstrate this practice. This is presented in section 2.2

4 *I note that the concept of "patchwork quilt" parameters is mentioned in section 2.1, again before any evidence (which is presented in Section 2.2 in Figure 1. This needs tidying up.*

The characterization as "Patchwork quilt" refers to the references provided in the sentence before. We revised the sentence to avoid this confusion.

5 *Also I was not clear at first how Point 2 was linked to Point 1. It seems like the paper starts with two unrelated paragraphs, and then later links them together using MPR in Point 4. This is not so easy to follow.*

We revised the text. A new sentence was introduced to bind points 1 and 2 so that the logic of the argumentation is straightforward.

[revised manuscript text omitted]